


# Modeling the contributions of global air temperature, synoptic-scale phenomena and soil moisture to near-surface static energy variability using artificial neural networks.

Sara C. Pryor[1], Ryan C. Sullivan[1] and Justin T. Schoof[2]

[1]Department of Earth and Atmospheric Sciences, Cornell University, Ithaca, NY 14853, USA.

[2]Department of Geography and Environmental Resource, Southern Illinois University, Carbondale, IL 62901, USA.

*Correspondence to:* S.C. Pryor (sp2279@cornell.edu)

**Abstract.** The static energy content of the atmosphere is increasing at the global scale, but exhibits important sub-
global and sub-regional scales of variability and is a useful parameter for integrating the net effect of changes in the

partitioning of energy at the surface and for improving understanding of the causes of so-called 'warming-holes'
(i.e. locations with decreasing daily maximum air temperatures ($T$) or increasing trends of lower magnitude than the
global mean). Further, measures of the static energy content (herein the equivalent potential temperature, $\theta_e$) are
more strongly linked to excess human mortality and morbidity than air temperature alone, and have great relevance
in understanding causes of past heat-related excess mortality and making projections of possible future events that

are likely to be associated with negative human health and economic consequences. A new non-linear statistical
model for summertime daily maximum and minimum $\theta_e$ is developed and used to advance understanding of drivers
of historical change and variability over the eastern USA. It is shown that soil moisture ($SM$) is particularly
important in determining the magnitude of $\theta_e$ over regions that have previously been identified as exhibiting
'warming holes' confirming the key importance of $SM$ in dictating the partitioning of net radiation into sensible and

latent heat and dictating trends in near-surface $T$ and $\theta_e$. Consistent with our *a priori* expectations, models built
using Artificial Neural Networks (ANN) out-perform linear models that do not permit interaction of the predictor
variables (global $T$, synoptic-scale meteorological conditions and $SM$). This is particularly marked in regions with
high variability in min- and max-$\theta_e$, where more complex models built using ANN with multiple hidden layers are
better able to capture the day-to-day variability in $\theta_e$ and the occurrence of extreme max-$\theta_e$. Over the entire domain

the ANN with 3 hidden layers exhibits high accuracy in predicting max-$\theta_e$ > 347 K. The median hit rate for max-$\theta_e$
> 347 K is > 0.60, while the median false alarm rate ≈ 0.08.

## 1    Motivation and objectives

Extreme heat is associated with significant societal and environmental impacts, and a number of prior studies have

elaborated the drivers of extreme air temperatures ($T$) and made projections of extreme $T$ and the associated human
health impacts and socioeconomic consequences (Sanderson and Ford, 2017;de'Donato et al., 2015;O'Neill and Ebi,
2009;Garcia-Herrera et al., 2010). Many previous studies have sought to quantify the predictability of extreme $T$ as




a function of lead-time and variables describing teleconnections to remote sea-surface temperature anomalies (McKinnon et al., 2016) and/or local soil moisture (Brabson et al., 2005). Physiological stress is maximized under the co-occurrence of elevated $T$ and specific humidity ($q$) (Zhang et al., 2014). Thus to understand spatiotemporal variability in heat-related mortality and/or morbidity there is a need to consider integrative variables derived from

both $T$ and $q$, such as equivalent potential temperature ($\theta_e$) computed herein using the following approximation:

$$\theta_e = T \left(\frac{1000}{P}\right)^{R_d/C_{pd}} + \frac{L(T)}{C_{pd}} Q, \tag{1}$$

Where $T$ is air temperature, $P$ is atmospheric pressure, $R_d$ is specific gas constant for air, $C_{pd}$ is specific heat for dry air, $L(T)$ is latent heat of vaporization ($f(T)$), $q$ is specific humidity of water vapor.

Equivalent potential temperature rather than equivalent temperature is used here to allow comparison of values from

other reanalyses (or other model output) that uses a different discretization of terrain elevation. While $T$ is responsive only to changes in the sensible heat flux from/to the surface, $\theta_e$ responds to changes in both the sensible and latent heat flux terms within the surface energy balance:

$$S(1 - \alpha) + R_{\mathrm{li}} - R_{\mathrm{lo}} = H + LE + G \tag{2}$$

Where $S$ is incoming solar radiation, $\alpha$ is albedo, $R_{li}$ is counter-radiated longwave radiation from the

atmosphere, $R_{lo}$ is outgoing longwave radiation from the surface, $H$ is sensible heat flux, $LE$ is latent heat flux, and $G$ is ground heat flux.

The total static energy of the global lower atmosphere (and thus $\theta_e$) has significantly increased since the 1970s due to the increase in net radiation (Peterson et al., 2011) and in response to increased $T$ and the associated response in $q$ (Ribera et al., 2004;Willett et al., 2007). However, climate modes (such as El Niño-Southern Oscillation (ENSO),

the North Atlantic Oscillation (NAO) and Pacific North American mode (PNA)) cause inter-annual variability in global (Huang et al., 2017) and regional (Llamedo et al., 2017) $T$ and $q$ anomalies, and there is substantial regional and sub-regional variability in absolute magnitude of static energy and temporal trends therein (Fall et al., 2010;Pryor and Schoof, 2016;Pryor et al., 2016).

The following is a précis of the scales and processes we seek to explore and include in our new statistical

downscaling model for summertime daily maximum and minimum equivalent potential temperature ($\theta_e$):

- Global scale forcing due to enhanced greenhouse gas concentrations and internal climate variability (e.g. ENSO (Mann et al., 1998)). As $T$ increases the atmospheric water vapor content responds in proportion to the saturation vapor pressure (Willett et al., 2007;Allen and Ingram, 2002). Thus, both components ($T$ and $q$) of static energy ($\theta_e$) are enhanced in years and seasons with high global mean air temperatures. Previous research

30        has indicated that variability in equivalent temperature ($T_e$, i.e. the temperature computed from (1) but excluding the correction for bringing the air parcel adiabatically to a reference pressure of 1000 hPa) in the North Atlantic is strongly linked to NAO (Ribera et al., 2004), and the probability of 'heat waves' across the US is linked to hemispheric waves (Teng et al., 2013) and thus the PNA (Trenberth, 1990).

- Synoptic scale forcing (Grotjahn et al., 2016). Much of the eastern USA broke records for heat indices during

35        the summer of 2012 in part due to persistent anticyclonic conditions (Peterson et al., 2013). Further, variability and temporal trends in summertime moist static energy and dew point temperatures ($T_d$) in the Midwestern USA





appear to be linked to enhanced horizontal atmospheric moisture advection due in part to northward expansion of the southeast summertime anticyclone (Kalkstein et al., 1998;Rogers et al., 2007;Pryor and Schoof, 2016;Ford and Schoof, 2017).

- Local-regional changes in water availability and energy partitioning at the surface, due to factors such as land cover change and soil moisture modification as a result of irrigation of cropland (Davey et al., 2006;Pryor et al., 2016).

Observed tendencies in $T$, $q$ and $\theta_e$ are naturally a product of a combination of these drivers (Horton et al., 2016). The expression of internal climates modes (e.g. ENSO, PNA and NAO) influences the frequency and intensity of different synoptic scale phenomena (Sheridan, 2003;Weaver, 2013), and has been found to be partly responsible for an increase in the number of oppressively hot days in many urban areas across the USA over recent decades (Sheridan et al., 2009). Further, there are important feedbacks between the land-atmosphere coupling, the synoptic scale circulation patterns and boundary-layer structure (Lee et al., 2016). Thus, for example, extreme $T_d$ (and by association elevated $q$) in the Midwestern USA is associated with (a) development and propagation of low pressure from the High Plains to through the upper Great Lakes, (b) healthy crops and sufficient surface soil moisture, and (c) restricted vertical mixing (Bentley and Stallins, 2008).

The objectives of this research are:

1) To use non-linear (machine learning) models applied to a three-tiered suite of predictors: (a) An index of the daily global mean temperature, (b) indices of the conditions at the synoptic scale based on principal components analysis (PCA) of upper-air variables, and (c) soil moisture estimates, to investigate spatiotemporal variations in $\theta_e$ and enhance mechanistic understanding of the causes of variability and change in $\theta_e$ over the eastern USA.

2) To examine whether the performance of the artificial neural networks (ANNs) used to predict daily maximum and minimum $\theta_e$ is enhanced by inclusion of additional hidden layers in the ANNs.

## 2    Data and Methods

### 2.1    Study region

The eastern USA (Figure 1a) is the focus of this research because:

- It is characterized by high summertime $T_e$ (and $\theta_e$) and in situ data have indicated trends in $T_e$ exceed those in $T$ alone (Schoof et al., 2015), and is largely congruent with an area of coherence for extreme $T$ events in prior research (McKinnon et al., 2016).

- It encompasses major urban areas that have experienced a number of past extreme heat events associated with substantial excess mortality and morbidity (see summary in (Vanos et al., 2015) and also (Anderson and Bell, 2011)). Further, a range of reanalyses exhibit a consistent signal of increasing frequency of both dry (i.e. high $T$ but low $q$) and humid (i.e. combined high $T$ and $q$) heat wave days over the study region (Schoof et al., 2017).

- It exhibits strong spatial gradients in terms of the nature of land cover and rate of change of both land management and soil moisture (Figure 1a and g, (Pryor et al., 2016;Pryor and Schoof, 2016;Ellenburg et al., 2016)). Parts of the region were identified in the global land–atmosphere coupling experiments (GLACE) as





exhibiting atypically strong atmosphere-surface coupling in some global climate models, and soil moisture makes a large contribution to sub-seasonal forecast skill for air temperatures and precipitation in this region (Koster et al., 2011;Koster et al., 2006).

- It also incorporates two areas of reduced daily maximum $T$ during parts or all of the twentieth century. One is located along the border of Iowa (IA)-Nebraska (NE)-South Dakota (SD) and one is centered on Mississippi (MS) and Alabama (AL) (locations shown in Figure 1h). The lack of warming in both regions has been attributed to changing land-surface characteristics and enhanced soil moisture availability (Kalnay and Cai, 2003;Pan et al., 2009;Ellenburg et al., 2016), and in the case of MS and AL upto 60% of the variance of summer temperatures has been ascribed to soil moisture (and thus an increase in the $LE$ at the surface at the expense of $H$, Eq. 2) and cloud cover (reducing the net radiation) (Ellenburg et al., 2016). However, additional factors may account for these regions of slow or absent warming including large scale climate modes such as the Interdecadal Pacific Oscillation and the associated sea surface temperature anomalies in the tropical Pacific (Meehl et al., 2015), and aerosol radiative forcing (Leibensperger et al., 2012;Yu et al., 2014).

## 2.2    Data set

All variables used herein derive from the MERRA-2 reanalysis data set (Molod et al., 2015;Reichle et al., 2017). This minimizes uncertainty in models linking our predictors to near-surface static energy resulting from use of different data sets to derive the predictor suite and/or the response variables (daily minimum and maximum $\theta_e$). The MERRA-2 reanalysis product assimilates an unprecedented array of remote sensing and in situ data streams, but does not assimilate in situ observations of near-surface $T$ or $q$. MERRA-2 output is available at a resolution of 0.625°× ~0.5° (longitude-by-latitude). We use MERRA-2 output for all summer days (JJA) during 1980-2015 to compute the following variables:

1) A globally averaged daily mean air temperature computed using hourly 2-m $T$ from all MERRA-2 grid cells. This predictor (Figure 1b) is intended to represent the long-term tendency in global mean temperatures and interannual/interdecadal variability caused by internal climate modes (Huang et al., 2017).

2) Indices of synoptic scale meteorology. Air temperature ($T_{850}$) and specific humidity ($q_{850}$) at 850 hPa along with 500 hPa geopotential heights ($Z_{500}$) in the domain (25.5-50°N, 97.5-65°W) for 20:00 UTC are converted to z-scores (that indicates how many standard deviations an individual value is from the mean) and used in a rotated PCA to generate daily principal component scores (PCs) that represent the proximity of each day to the major modes of synoptic-scale variability. Fifteen components are retained based on a scree plot analysis (Cattell, 1966) and are subject to Varimax rotation (Richman, 1986). Daily PC scores for all 15 PCs are used as predictors in the statistical models, to allow each day to exhibit partial membership of multiple synoptic types. Spatial fields of the three variables sampled once per day are used in the PCA due to the high temporal autocorrelation present in these variables, and 20:00 UTC is selected to coincide with approximately the timing of the afternoon peak in surface $T$ over the eastern USA. Figure 2 shows centroids of synoptic modes of variability as defined using the PCA (so-called key days) as represented by



the spatial patterns of $T_{850}$, $q_{850}$ and $Z_{500}$ computed as the mean conditions on the seven days that exhibit highest PC scores on each PC. As shown, many of the synoptic types thus identified are readily interpretable as representing a diversity of zonal versus meridional circulation (cf. type 9 and 5 and 6), and some are characterized by conditions known to be associated with strong low-level advection of $T$ and $q$

into the region (e.g. types 1, 3, and 15) (Pryor and Schoof, 2016;Weaver, 2013). Further, most types exhibit a high degree of similarity with other synoptic-scale classifications derived for the region (e.g. type 7 is very similar to one that is associated with summertime precipitation over the southeastern USA (Diem, 2006)).

3) An index of soil moisture (*SM*). Following previous research (Ford and Schoof, 2016), we use a time and

spatially integrated metric of *SM*. Specifically we use a 90-day running mean estimate of antecedent *SM* in 3×3 grid cells centered on the grid cell in question (see Figure 1g). MERRA-2 *SM* has been extensively evaluated relative to in situ measurements of surface and root-zone *SM* and exhibits an unbiased root mean square error (RMSE) of 0.05 $m^3 m^{-3}$ and a variance explanation ($R^2$) value of the average root-zone *SM* anomaly of 0.56 (Reichle et al., 2017).

4) An estimate of (i) daily maximum $\theta_e$ and (ii) daily minimum $\theta_e$ in each grid cell (see Figure 1c and d) derived using Eq(1) along with hourly $T$ at 2-m, $q$ at 2-m and surface pressure ($P$). The domain used to compute the gridded fields of the predictands (daily minimum and maximum $\theta_e$) is truncated by one grid cell on each boundary of the domain used to generate the predictors to accommodate the spatial averaging used to generate the *SM* predictor. The range of grid cell $\theta_e$ estimates are consistent with those derived from

station observations within the study region (Pryor and Schoof, 2016), and in accord with *a priori* expectations both daily max-$\theta_e$ and min-$\theta_e$ exhibit primarily latitudinal variability (Figure 1c and d). Mean max-$\theta_e$ decreases from values of ~350 K (~80°C) in the south of the domain to approximately 320 K (~50°C) in the north. MO and IA exhibit anomalously high mean max-$\theta_e$ compared with grid cells at the same latitudes (Figure 1c), reflecting the advection of air with high $T$ and $q$ from the south by the Great

Plains Low Level Jet (GPLLJ) (Weaver, 2013;Pryor and Schoof, 2016;Schoof et al., 2015), and the presence of abundant *SM* (Figure 1g). There are also important spatial patterns of the day-to-day variability of daily max-$\theta_e$ and min-$\theta_e$ that provide key context for considering the performance of different transfer functions (i.e. statistical models linking indices of the global mean T, synoptic scale meteorology and soil moisture (the predictors) to the response variables; max-$\theta_e$ and min-$\theta_e$). The variance (and standard

deviation) of daily max-$\theta_e$ values (i.e. a measure of the dispersion of individual days around the mean values of min- and max-$\theta_e$ shown in Figures 1c and 1d) is largest over IA/southern Minnesota (MN) (Figure 1e), while the variance of min-$\theta_e$ is greatest over Illinois (IL) (Figure 1f). In both cases the day-to-day variability as measured by the standard deviation is ~ 10 K. The standard deviation of daily max-$\theta_e$ and min-$\theta_e$ decreases with decreasing latitude and both variables exhibit lowest variability over he southern

portions of the study domain (e.g. Florida (FL) has a standard deviation < 5 K) (Figure 1e,f).

Thus the time series of predictors 1) and 2) (global mean $T$ and the 15 PC scores) are common to models built for all





grid cells, but predictor 3) (*SM*) and the response variables (predictands, daily max-$\theta_e$ and min-$\theta_e$) are grid-cell specific.

### 2.3    Methods

Artificial neural network (ANN) architectures are potentially highly useful in developing statistical models for
response variables such as $\theta_e$ because ANN do not require any assumptions about the form of the relationship between individual predictors, between predictors and predictands (min- and max-$\theta_e$), and can treat complex and non-linear term interactions. ANNs are data-driven self-adaptive multi-layer perceptrons that model relationships between input variables and dependent output variables. Term interactions are described using mathematical functions encoded within hidden-layers and weights that connect all nodes within the network layers including the
input (predictors) and output (predictand) layers to perform the non-linear mapping between the input and output variables (Gardner and Dorling, 1998). The number of hidden layers within the ANN determines the degree of non-linearity that can be modeled. Hence, if the data are linearly separable, no hidden layers are required. Our *a priori* expectation is that the different predictors of daily maximum and minimum $\theta_e$ will interact in complex, non-linear ways. Thus, we apply ANN to develop models relating the global mean *T*, PC scores of the synoptic-scale
meteorology and antecedent *SM* to daily maximum or minimum $\theta_e$ in each grid cell. Because we seek to examine spatial variability in model performance, we build and test the ANNs at the grid-cell level and then examine the resulting spatial coherence of model skill. A range of different learning algorithms can be employed in ANN. Herein the neural networks are constructed within Matlab using the Levenberg-Marquardt back-propagation algorithm (in which the sum of the squares of the deviations between the observations and model predictions is minimized)
(Papageorgiou and Poczeta, 2017). Although there is no single 'best-practice' regarding the number of hidden layers to use with ANN, there is evidence that a single hidden layer is sufficient for the large majority of problems (Toth et al., 2000). To test the dependence of model skill on the number of hidden-layers, three independent models are constructed for each MERRA-2 grid cell using:

    1)   No hidden layers, i.e. a linear regression model with no interaction between the predictors.
25       2)   ANN with a single hidden layer.
    3)   ANN with three hidden layers.

To examine the importance of *SM* in determining the downscaling model skill, a fourth ANN model (with 3 hidden layers) is also built for each grid cell and each of the two predictands (daily maximum and minimum $\theta_e$) that excludes *SM* from the input variables. Lastly, it is challenging to determine which measures of *SM* are most
appropriate to use within statistical downscaling models. Therefore, in addition to developing models using the MERRA-2 variable 'PRMC', which is the 'Total profile soil moisture content' in $m^3m^{-3}$ (and is summed across all six soil layers), a fifth ANN model (with 3 hidden layers) is also built that uses the variable 'GWETTOP' that describes the *SM* content in the upper 5 cm of the soil (unitless) (Reichle et al., 2017), and thus best represents the *SM* that is readily available to the overlying atmosphere.

Table 1 summarizes the model abbreviations used herein, and a schematic of the downscaling model architecture and data flows is given in Figure 3. For each model (and thus each grid cell) 70% of the data set is randomly



selected to be used for training of the model, 15% is used for internal validation and 15% is withheld and used as an independent sample for model testing. We use two primary metrics of model performance: RMSE and Pearson correlation coefficient (r) between observed and predicted daily minimum and maximum-$\theta_e$ in each grid cell on each day in the test data set (i.e. independent observations), summarized at both the grid cell level and also averaged over

all 1962 MERRA-2 grid cells that have some land areas within them (i.e. a domain average). The correlation coefficient is thus used as a relative measure of model performance, while RMSE provides an absolute measure of degree of agreement between the model 'predictions' and the observed values (i.e. it is the typical value of the prediction error). Given the importance of extreme heat to human health we further examine the ability of the models to capture the occurrence of very high $\theta_e$. In this analysis we set a threshold of 347 K (73°C) to indicate

extreme max-$\theta_e$ (based on information provided in (Buzan et al., 2015) for the eastern USA) and a threshold of 337 K for min-$\theta_e$. A contingency table approach is used to evaluate the accuracy of the model predictions of extreme-$\theta_e$ using the hit rate (*HR*) (Wilks, 2011):

$$HR = \frac{\#hits}{\#hits + \#misses} \quad , \tag{3}$$

Where *#hits* is number of days in the independent data set where the observations and predictions both indicate

exceedance of the threshold, and *#misses* is the number of days when the observations indicate an exceedance of the threshold but the model prediction does not.

And the false alarm rate (*FAR*) of each individual grid cell:

$$FAR = \frac{\#false\ alarms}{\#false\ alarms + \#correct\ non-events} \quad , \tag{4}$$

Where *#false alarms* is number of days in the independent data set where the observations did not indicate

exceedance of the threshold but the prediction was for an exceedance, and *#correct non-events* is the number of days when the observations and predictions both indicate the threshold is not exceeded.

## 3    Results

For both daily maximum-$\theta_e$ and daily minimum-$\theta_e$ the worst model performance statistics (highest RMSE as a fraction of the temporal variability $\theta_e$, and lowest *r*) are associated with the linear models that do not include

parameter interactions (i.e. MLR) (Table 1 and Figures 4 and 5). Nevertheless, output from all model architectures for min-$\theta_e$ and max-$\theta_e$ exhibit high correlation coefficients (r > 0.8) with independent data over most of the study domain. Correlation coefficients exceed 0.8 for 84% of grid cells for ANN-HL3-SM applied to max-$\theta_e$ and 81% for min-$\theta_e$ and RMSE < 5 K (92% of grid cells for ANN-HL3-SM exhibit a RMSE below this threshold for max-$\theta_e$ and 91% for min-$\theta_e$) (Table 1, Figure 4 and 5). Grid cells with highest RMSE for min-$\theta_e$ and max-$\theta_e$ also generally have

highest variance (i.e. largest day-to-day variability). For example, grid cells in IA exhibit highest variance and highest RMSE for max-$\theta_e$ (Figure 1e and Figure 4), and grid-cells within IL are generally characterized by large RMSE and variance of min-$\theta_e$ (Figure 1f and Figure 5). Thus, while noting the RMSE (i.e. typical prediction error) is largest over IA for max-$\theta_e$ (~ 5 K), it is less than half the standard deviation computed from the day-to-day variability in max-$\theta_e$ (~ 10 K).



Generally all models exhibit slightly worse performance across both measures (r and RMSE) for min-$\theta_e$ than max-$\theta_e$ at the grid-cell level and integrated over all land grid cells (Table 1 and Figures 4 and 5). The reduced model skill for min-$\theta_e$ may reflect use of output at 20:00 UTC values of the predictors used in the synoptic classification due to our particular focus on daytime max-$\theta_e$.

Although performance differences between the five model architectures for daily maximum-$\theta_e$ and daily minimum-$\theta_e$ are comparatively modest when averaged over the entire domain (Table 1), there are important regional variations in the performance of the different model functional forms. Over two-thirds of all grid cells (1332 of 1962) exhibit lower RMSE in the ANN model with 3 hidden layers and including *SM* (i.e. ANN-HL3-SM) than in any of the other models (e.g. MLR). The enhancement of model performance as measured by a decrease in RMSE for the more

complex model of max-$\theta_e$ and min-$\theta_e$ is particularly marked in the west-central of the domain (over parts of Missouri (MO) and Iowa (IA), close to or within one of the 'warming holes') (Figure 4 and 5). This is a region where a substantial fraction of *T* variance is explained by thermal and moisture advection by the GPLLJ (Weaver, 2013), where land-atmosphere coupling is particularly intense (Koster et al., 2011;Koster et al., 2006) and where there are strong horizontal gradients of *SM* (Figure 1g). Lowest correlations between predicted and observed min-

and max-$\theta_e$ values occur over east Texas (TX) for all model formulations although the RMSE of model predictions are not particularly high in this area (Figure 4 and 5). The low RMSE may reflect the small day-to-day variability in min- and max-$\theta_e$ over this region (Figure 1e and f), possibly due to the proximity to the ocean, while the low r may indicate that the synoptic types derived herein are not able to represent mesoscale features such dry lines that play a key role in dictating day-to-day variability in $\theta_e$ over this sub-region. It is also worthy of note that this area was

excluded from the eastern US in terms of the area of coherence for extreme *T* over the eastern USA (McKinnon et al., 2016) and that eastern TX is on the southwestern boundary of the study domain. These two factors may indicate that the synoptic types derived herein do not fully represent the range of meteorological conditions associated with $\theta_e$ variability.

Interpreting weights from complex ANN is very challenging in the context of predictor relevance and the values

(and sign) of the weights vary in space and with the complexity of the model architecture (number of hidden layers). However, it is worthy of note that the weights on the index of global temperatures (and thus expression of internal climate modes) in ANN-HL3-SM are uniformly close to zero across the grid-cells in the interior of the continent, but are of large magnitude in land grid cells close to the Gulf of Mexico (i.e. the south of the domain) and around the Great Lakes. Given this index is strongly influenced by global sea surface temperatures (SST), the implication is

that this predictor contains important information about the SST and thus evaporation from the Great Lakes and the Gulf of Mexico.

Consistent with prior research that has indicated the importance of atmosphere-surface interactions (Cai et al., 2014) and specifically soil moisture (Pryor et al., 2016;Seneviratne et al., 2010) in surface energy partitioning and thus near-surface *T* and *q* regimes and static energy, exclusion of *SM* from the ANN with 3 hidden layers (i.e. ANN-

HL3) decreases model performance relative to ANN-HL3-SM and increases the RMSE for max-$\theta_e$ in 70% of grid cells. The regions for which this impact is most strongly manifest are close to or within the 'warming holes' described above and/or are located downstream of regions of significant county-level irrigation and anthropogenic



enhancement of *SM* (Pryor et al., 2016) (Figure 1g). For example, RMSE for max-$\theta_e$ is increased in models excluding *SM* in all grid cells within MO, and all but one grid cell each in IA and IL (Figure 4). This finding is also replicated in the second region of weak or negative air temperature trends described above and centered on MS and AL (Ellenburg et al., 2016). The RMSE is lower in ANN-HL3-SM than ANN-HL3 over all but one grid cell in these

two states. Thus this analysis strongly supports prior assertions that *SM* plays a key role in dictating the surface energy balance and in the suppression of daily maximum *T*, while increasing max-$\theta_e$.

The statistical (downscaling) models show similar dynamic range to independent observations, although there is some evidence that the models underestimate the total variance in max-$\theta_e$ leading to underestimation of extreme max-$\theta_e$ as is evident from the flattening of the scatterplots for very high values of daily maximum $\theta_e$ (see the upper

row of panels in Figure 4). To examine this further we conduct an analysis of the *HR* and *FAR* for max-$\theta_e$ in excess of 347 K. This threshold is exceeded by daily maximum $\theta_e$ derived from the MERRA-2 reanalysis on an average of ~ 15% of summer days when all grid-cells are considered, but naturally exhibits a higher frequency of exceedance (of up to 75% of days) along the southeastern portion of the TX gulf coast and is observed on nearly 50% of days over coastal portions of the Gulf coast states and FL (Figure 6a). Conversely, it is seldom or never observed within

grid cells in the north of the domain (Figure 6a). To ensure a sufficiently robust sample size on which to compute the *HR* for extreme max-$\theta_e$ we consider only grid cells where more than 40 days in the test period (i.e. 8%) exceed this threshold. The mean *HR* values for the linear model (MLR), the ANN with 3-hidden layers (ANN-HL3-SM), and the ANN with 3-hidden layers but excluding SM (ANN-HL3) computed over all these grid cells is 59, 60 and 56%, respectively (Figure 6b-d), indicating that over the entire study domain the role of SM and predictor

interactions in explaining the occurrence of extreme max-$\theta_e$ is modest. All model forms perform least well in terms of predicting the occurrence of max-$\theta_e$ > 347 K over eastern TX and South Carolina (SC) (Figure 6e-g). However, the model excluding *SM* exhibits particularly poor performance (low *HR*) in these regions. The causes of the poor model performance in eastern TX and SC are currently not fully understood, although it is worthy of note that data from MERRA-2 grid-cells in SC exhibit a relatively low overall frequency of exceedance of this threshold and are

also characterized by comparatively low 99[th] percentile $\theta_e$ in an analysis of heat indices derived from the Community Land Model v4.5 (Buzan et al., 2015). Grid cells along the Gulf coast and over the states of MO, IA and IL exhibit high *HR* for prediction of extreme max-$\theta_e$ and substantial improvement in *HR* is noted in IA, IL and MO (Figure 6e-g) in the ANN-HL3-SM relative to the other model forms. This is consistent with strong spatial gradients in *SM* (Figure 1e), findings of the GLACE projects of strong atmosphere-surface coupling (Koster et al.,

2011;Koster et al., 2006), and analyses for stations in IL that also show a strong dependence of high $T_e$ on soil moisture (Ford and Schoof, 2016). To contextualize the *HR* presented above it is important to note that they are associated with comparatively low false alarm rates (*FAR*). Indeed, *FAR* for the occurrence of min-$\theta_e$ > 342 K or max-$\theta_e$ > 347 K are very modest for all model formulations (Figure 6 and 7). For example, over 94% of grid cells indicate *FAR* for max-$\theta_e$ > 347 K that are below 0.25 for the ANN-HL3-SM models. Thus, the relatively high *HR*

reported herein are not being artificially inflated by unrealistically high predictions of the occurrence of extreme $\theta_e$.

The inclusion of *SM* as a predictor greatly enhances *HR* in regions previously identified as exhibiting high variance





in extreme $\theta_e$ without a concomitant increase in *FAR* (Figure 6 and 7). It should be acknowledged that even the ANN with 3 hidden layers and soil moisture (ANN-HL3-SM) exhibits a modal grid-cell *HR* of 0.6-0.7, and thus misses a substantial fraction of extreme $\theta_e$. Nevertheless, these *HR* and *FAR* are indicative of positive Relative Operating Characteristics (ROC) (i.e. plots of the true positive rate greatly exceeds false positive rates) (Wilks, 2011). Further, *HR* and *FAR* are comparable to (or better than) seasonal re-forecasts of summertime *T* at 2-m over the land areas of Southern Europe developed using the European Centre for Medium-Range Weather Forecasts (ECMWF) seasonal ensemble forecasting system (Weisheimer et al., 2011).

In contrast to the results for prediction of extreme max-$\theta_e$ the model architecture has virtually no impact on *HR* for min-$\theta_e > 342$ K, and neither does the inclusion of *SM* in the model. In all cases the domain averaged *HR* = 59% and no region exhibits consistent improved or degraded performance for ANN-HL3-SM or ANN-HL3 over MLR (Figure 7). This finding is consistent with the overall results for models of min-$\theta_e$ that exhibit only modest decreases in model performance (increased RMSE and lower r) when *SM* is excluded from the predictor suite (Table 1 and Figure 5). Consistent with the interpretation of the surface energy balance (Eq 2) this re-emphasizes that *SM* more directly impacts near-surface *T* and *q* during the daytime under conditions of positive net radiation.

Differences in model performance between ANN conditioned on total *SM* and using wetness only in the top soil layer (upper 5 cm) are very small when averaged across the domain (Table 1) and indeed for virtually all grid cells. Only 26 grid cells exhibited a Δ|RMSE| > 0.5 K for models using PRMC versus those using GWETTOP (out of a total of 1962), while 155 exhibited an increase in RMSE > 0.5 K when *SM* was excluded from the model. Thus, although the weights within the ANNs differ for use of the two *SM* parameters, the overall model skill is unchanged by use of the two *SM* estimates possibly due to the spatial and temporal averaging applied herein, or uncertainty in reanalysis-derived *SM* variables.

## 4    Summary and Conclusions

Very few statistical downscaling analyses focus on integrative variables such as $\theta_e$ that explicitly incorporate co-variability of *T* and *q*, but such variables have direct applications to climate change impact analyses (such as analysis of heat waves (Buzan et al., 2015)). Further, this is an application of climate downscaling where statistical approaches may be particularly useful given evidence that even when nested within observed lateral boundary conditions Regional Climate Models (RCMs) have difficulty in capturing the joint probability distributions of *T* and *q* and thus in accurately representing either the probability distribution of static energy or the spatio-temporal variability therein (Pryor and Schoof, 2016). Analyses of $\theta_e$ are also essential to advancing fundamental understanding of changes in the total static energy content of the lower atmosphere, and may reveal important information of relevance to both model performance analyses and attribution studies of global change.

The goal of this work is to develop a hierarchy of statistical models with increasing complexity and use them to determine the degree to which increased complexity enhances the skill of model predictions of $\theta_e$ and to attribute variability in min- and max-$\theta_e$ over eastern North America. Prior to discussing the results from application of this analysis framework to output from the EMRRA-2 reanalysis it is worthy of note that previous research on regional



heat wave characteristics over the contiguous US using a suite of reanalyses indicated some important differences in the magnitude of derived equivalent temperature ($T_e$) between the reanalysis products (Schoof et al., 2017). Thus, there would be value in applying this framework to additional observationally constrained data sets to evaluate the degree to which the findings of a key role of $SM$ to determining the model skill for daily maximum $\theta_e$ in specific
sub-regions are generalizable and spatially consistent between reanalyses and equally to determine if the partial truncation of the upper percentiles of daily maximum $\theta_e$ in the model predictions is also a generalizable finding when our model framework is applied to different data sets.

Consistent with our *a priori* expectations, models built using ANN out-perform those that do not permit interaction of the predictor variables. Domain averaged RMSE for min- and max-$\theta_e$ is smallest in the more complex models
(i.e. for an ANN with 3 hidden layers, RMSE < 4 K and < 4.3 K, respectively, c.f. mean max-$\theta_e \approx 333$ K and mean min-$\theta_e \approx 321$ K). Particularly in regions with high variability in min- and max-$\theta_e$ the more complex models with multiple hidden layers are better able to capture the day-to-day variability in $\theta_e$. Correlation coefficients exceed 0.8 for 84% of grid cells for ANN-HL3-SM applied to max-$\theta_e$ and 81% for min-$\theta_e$ and 92% of grid cells for ANN-HL3-SM exhibit a RMSE < 5 K for max-$\theta_e$ and 91% for min-$\theta_e$. This is also true for the simulation of extreme min- and
max-$\theta_e$. The ANN with 3 hidden layers and that includes $SM$ as a predictor exhibits a domain averaged median hit rate for max-$\theta_e$ > 347 K is > 0.60, while the median FAR $\approx 0.08$. Results from the ANN models further indicate that max-$\theta_e$ and the occurrence of extreme max-$\theta_e$ appear to be considerably more sensitive to $SM$ than min-$\theta_e$ which in turn exhibits a stronger dependence on the precise prevailing synoptic scale conditions.

We can not conclusively discount contributions from other phenomena to the occurrence of 'warming holes' (areas
with declining or no-trends in $T$) (Meehl et al., 2015), or that these features are a complex response to multiple drivers. However, results presented herein are consistent with past work that has indicated the importance of soil moisture ($SM$) in determining partitioning of the surface energy budget, and thus the spatiotemporal patterns of $\theta_e$ over the eastern USA (Koster et al., 2011;Koster et al., 2006;Pryor and Schoof, 2016;Pryor et al., 2016;Ford and Schoof, 2016, 2017;McKinnon et al., 2016). Indeed, $SM$ is particularly important in determining the surface energy
partitioning and the magnitude of $\theta_e$ over regions that have previously been identified as exhibiting 'warming holes', and for all grid cells the RMSE for models including $SM$ as a predictor is smaller than the temporal variability of $\theta_e$ as measured using the standard deviation of the daily $\theta_e$ values. Specifically, only a model including $SM$ is able to predict the occurrence of extreme (and highly health-relevant) values of $\theta_e$ over the western portion of Midwestern states such as IA, MO, IL and also in MS and AL. This research thus implies that $SM$ has played and may continue
to play a key role in dictating the presence and intensity of 'warming holes' that have been previously noted in analyses of near-surface air temperature data (from both in situ measurements and reanalysis products).

The primary purposes of this research are to enhance mechanistic understanding of the causes of variability and change in $\theta_e$ over the eastern USA and to propose a new downscaling approach to allow projections of daily minimum and maximum $\theta_e$ using variables commonly available from reanalyses and global and regional climate
models. However, these analyses also may have applications to short-term forecasting of human-health relevant heat events (McKinnon et al., 2016;Weisheimer et al., 2011), since the methodological framework developed herein




could be applied to observed antecedent *SM*, and modeled forecasts of the global mean *T* and conditions at the synoptic scale over the eastern USA.

## 5    Data availability

The MERRA-2 data used herein are available from http://disc.sci.gsfc.nasa.gov/daac-bin/FTPSubset2.pl (see
(Bosilovich et al., 2015) for a detailed description of the file structures).

## 6    Acknowledgments

The authors gratefully acknowledge funding from the US National Science Foundation under grants 1339629, 1339655 and 1502400, the US Department of Energy (DE-SC0016438) and the US National Aeronautics and Space Administration (14-EARTH14F-0207). Any opinions, findings, and conclusions or recommendations expressed in
this material are those of the authors and do not necessarily reflect the view of the National Science Foundation, National Aeronautics and Space Administration or Department of Energy.

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



**Table 1. Domain averaged model performance statistics (root mean square error (RMSE) and Pearson correlation coefficient (r)) versus independent test data for daily maximum equivalent potential temperature (max-$\theta_e$) and daily minimum equivalent potential temperature (min-$\theta_e$). The results are shown for the five model formulations expressed in terms of the number of hidden layers in the ANN (where 0 hidden layers indicates results for a multiple linear regression model with no term interactions), the presence of absence of soil moisture as a predictor and whether the *SM* is total columnar (PRMC) or surface only (WGETTOP).**

| Abbreviation used herein | Model architecture | Soil moisture predictor | max-$\theta_e$ | | | min-$\theta_e$ | | |
|---|---|---|---|---|---|---|---|---|
| | | | \<RMSE\> (K) | \<r\> | # grid cells with r>0.8 & RMSE < 5K | \<RMSE\> (K) | \<r\> | # grid cells with r>0.8 & RMSE < 5K |
| MLR | Multiple linear regression | PRMC | 4.05 | 0.849 | 1575, 1727 | 4.39 | 0.836 | 1537, 1665 |
| ANN-HL1-SM | ANN with 1 hidden layer | PRMC | 4.03 | 0.850 | 1602, 1754 | 4.34 | 0.839 | 1572, 1750 |
| ANN-HL3-SM | ANN with 3 hidden layers | PRMC | 3.97 | 0.855 | 1621, 1798 | 4.29 | 0.844 | 1588, 1782 |
| ANN-HL3 | ANN with 3 hidden layers, but excluding *SM* | None | 4.10 | 0.846 | 1549, 1675 | 4.33 | 0.843 | 1580, 1765 |
| ANN-HL3-TOP | ANN with 3 hidden layers | WGETTOP | 3.97 | 0.856 | 1623, 1802 | 4.29 | 0.844 | 1604, 1782 |



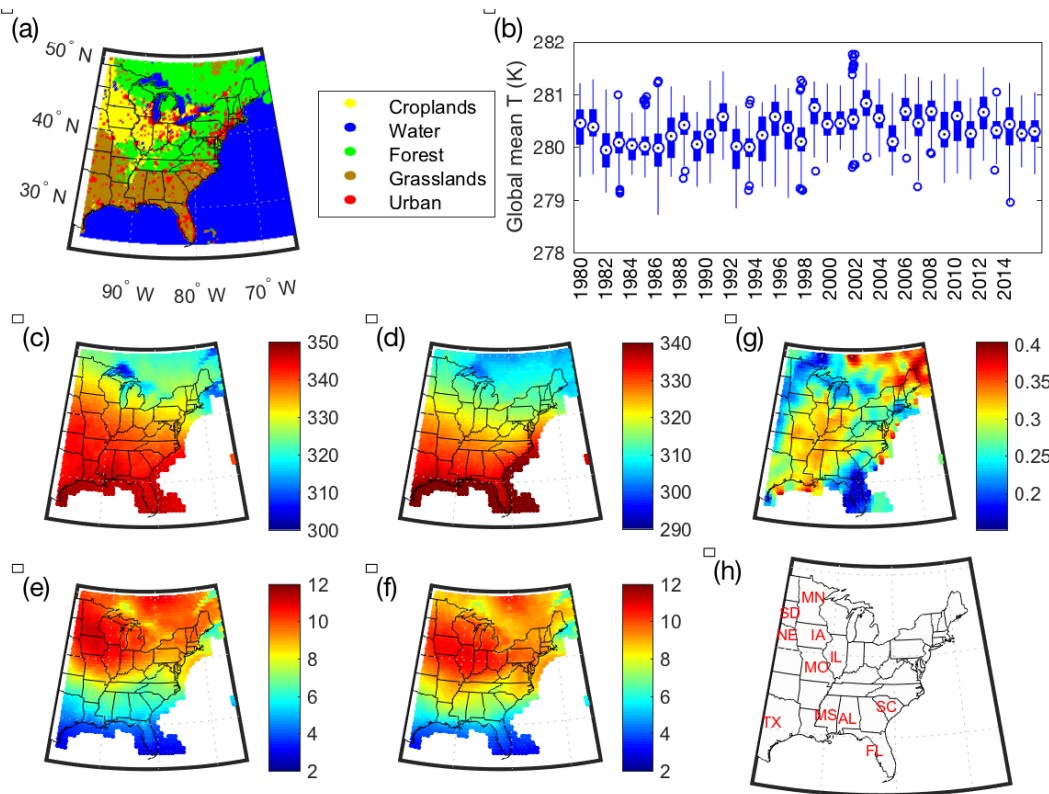

**Figure 1. (a) The study domain used herein and five consolidated land use land-cover (LULC) classes as represented at a resolution of 0.05×0.05° from the MODIS land cover data set for 2014 (MCD12C). (b) A boxplot of daily global mean JJA temperature values (K) (1980-2014) as computed from the MERRA-2 output. (c) Mean summertime (JJA) maximum-$\theta_e$ and (d) minimum-$\theta_e$ (K) as computed from hourly $T$ and $q$ at 2-m and surface pressure ($P$) in each MERRA-2 grid cell. Standard deviation of daily (e) maximum-$\theta_e$ and (f) minimum-$\theta_e$ (K). (g) Mean soil moisture estimates from MERRA-2 used as a predictor in the ANN and computed as follows: the daily mean value for total profile soil moisture content (PRMC) for each grid cell is averaged over a spatial area of 3×3 grid cells centered on the grid cell of interest, and used to compute a 90-day running JJA mean soil moisture (m$^3$m$^{-3}$). Panel (h) shows the location of states that are explicitly referred to herein.**





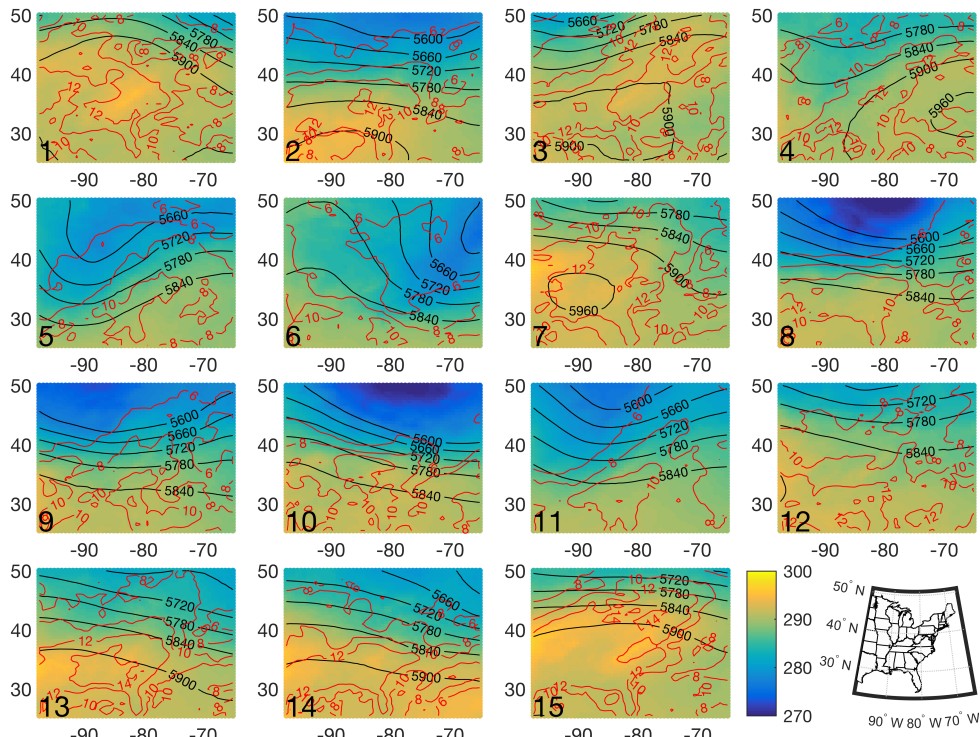

**Figure 2. Summary of the conditions under each of the PCA-derived synoptic modes of variability. The panels show the "mode centroid" (i.e. mean of the 7 days that exhibit highest PC scores for the type). The color depicts $T_{850}$ (K), the solid black lines show $Z_{500}$ (gpm) and the red lines depict $q_{850}$ (gKg$^{-1}$). The PC number is shown in the lower left of each panel and the panels are arranged so that the first mode of variability is shown in the upper left panel, the fifth PC is shown in the second row in the left most panel and so forth. To aid legibility, the panels in this figure denote the study region as a rectangle, with state outlines excluded, thus also shown in the lower right of this figure is a map depicting the state outlines for the same domain as shown in the other panels but using a Plate Carree projection (i.e. the equi-rectangular projection) used in Figures 1, 4, 5, 6 and 7.**



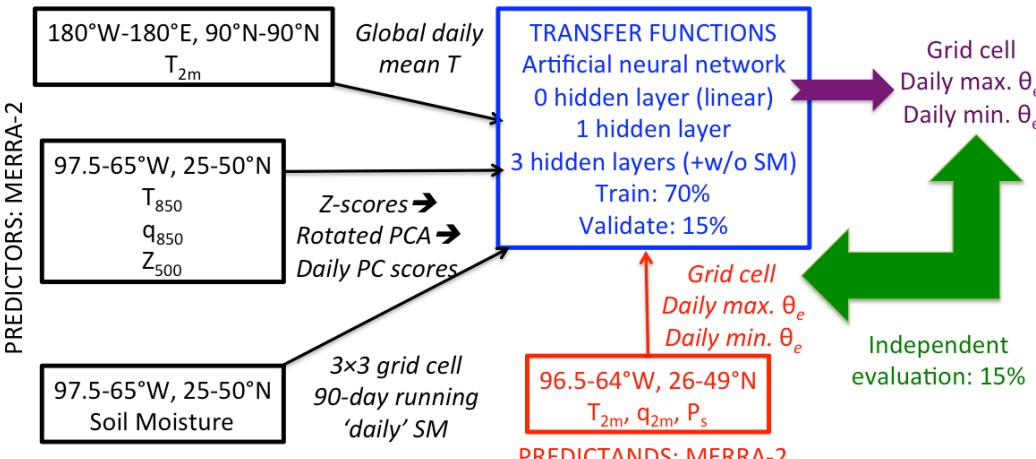

**Figure 3. Schematic of the ANN model architecture, data flow, predictors and predictands.**





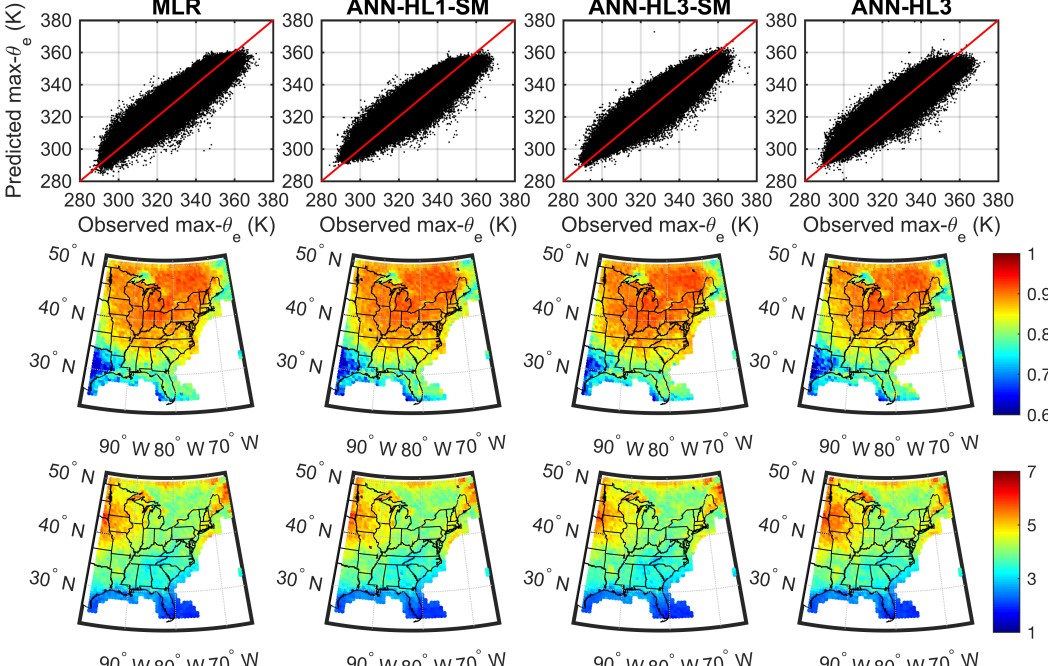

**Figure 4. Précis of the downscaling model performance for daily maximum-$\theta_e$ (max-$\theta_e$). The upper panels show scatterplots of observed and predicted max-$\theta_e$ for all grid cells and all days within the test (independent) data, along with a red 1:1 line. The columns show results for four model configurations: multiple linear regression (MLR, left), ANN with 1-hidden layer (ANN-HL1-SM, second column), ANN with 3-hidden layers (ANN-HL3-SM third column) and ANN with 3-hidden layers but excluding information regarding the soil moisture (ANN-HL3 right column). The second row shows the Pearson correlation (r) of predicted and observed max-$\theta_e$ values in the independent data for the four different models, while the third row shows the mean root mean square error (RMSE) (K) computed for independent data from each grid cell for the four models.**





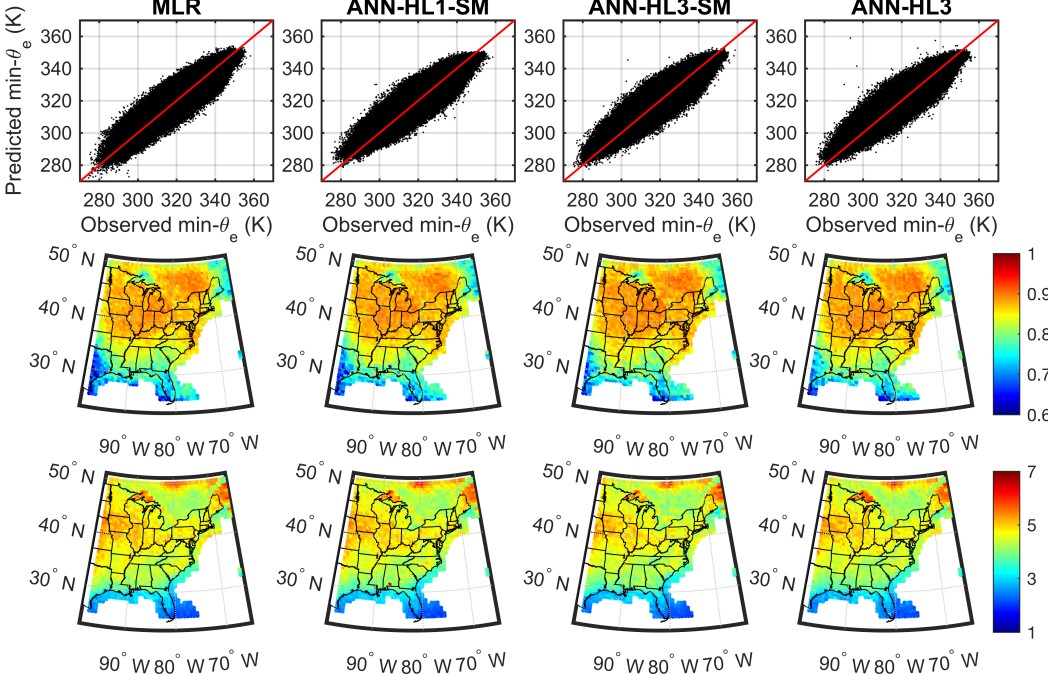

**Figure 5. Précis of the downscaling model performance for daily minimum-$\theta_e$ (min-$\theta_e$). The upper panels show scatterplots of observed and predicted min-$\theta_e$ for all grid cells and all days within the test (independent) data, along with a red 1:1 line. The columns show results for four model configurations: multiple linear regression (MLR, left), ANN with 1-hidden layer (ANN-HL1-SM, second column), ANN with 3-hidden layers (ANN-HL3-SM third column) and ANN with 3-hidden layers but excluding information regarding the soil moisture (ANN-HL3 right column). The second row shows the Pearson correlation (r) of predicted and observed min-$\theta_e$ values in the independent data for the four different models, while the third row shows the mean root mean square error (RMSE) (K) computed for independent data from each grid cell for the four models.**



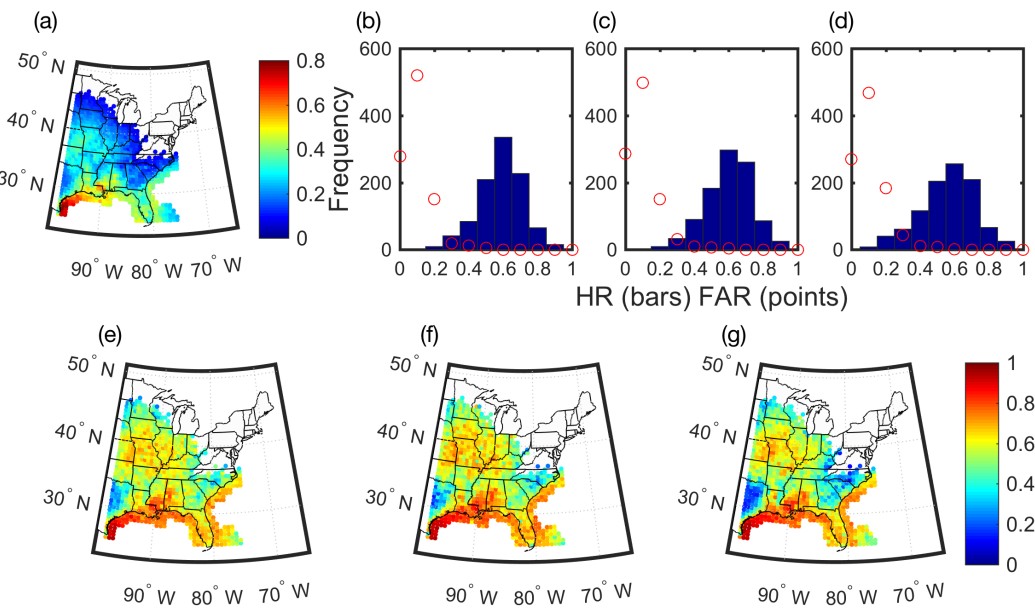

Figure 6. (a) Map of the fractional frequency of occurrence of max-$\theta_e$ > 347 K in each grid cell based on the independent (test) data set. (b) Histogram of hit rate (*HR*) and false alarm rate (*FAR*) for all grid cells and (e) spatial map for correct identification of max-$\theta_e$>347 K for all grid cells where this threshold is exceeded by >8% of days in the independent data based on output from the multiple linear regression model (MLR). (c) and (f) As in panels (b) and (e) but for output from the ANN model with 3 hidden layers and including *SM* (ANN-HL3-SM). (d) and (g) As in panels (b) and (e) but for output from the ANN model with 3 hidden layers excluding *SM* (ANN-HL3).



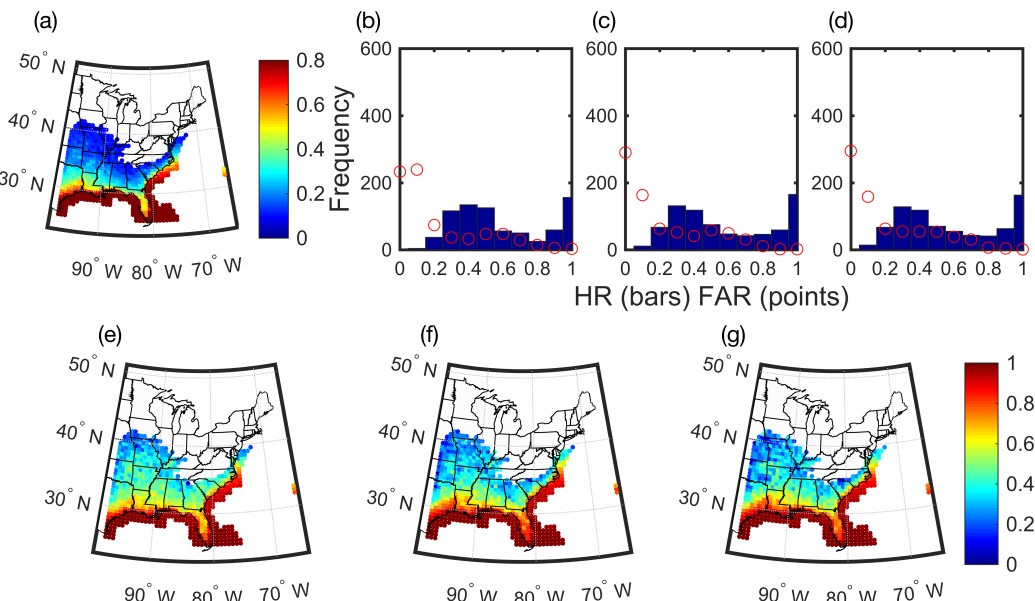

**Figure 7. (a) Map of the fractional frequency of occurrence of min-$\theta_e$ > 337 K in each grid cell based on the independent (test) data set. (b) Histogram of hit rate (*HR*) and false alarm rate (*FAR*) for all grid cells and (e) spatial map for correct identification of min-$\theta_e$>337 K for all grid cells where this threshold is exceeded by >8% of days in the independent data based on output from the multiple linear regression model (MLR). (c) and (f) As in panels (b) and (e) but for output from the ANN model with 3 hidden layers and including *SM* (ANN-HL3-SM). (d) and (g) As in panels (b) and (e) but for output from the ANN model with 3 hidden layers excluding *SM* (ANN-HL3).**

