# Peer review of "Modeling the contributions of global air temperature, synoptic-scale phenomena and soil moisture to near-surface static energy variability using artificial neural networks."

_Atmospheric Chemistry and Physics, 2017_

## Referee Comment (RC1) · Anonymous Referee #2 · 30 Jul 2017

The following is a review of the article entitled "Modeling the contributions of global air temperature, synoptic-scale phenomena and soil moisture to near-surface static energy variability over the eastern U.S.A" by Pryor et al., submitted to Atmospheric Chemistry and Physics. The article details the application of five candidate ANN models of varying complexity in the prediction of daily equivalent potential temperature over the eastern U.S. during the summer (JJA). The ANNs each use inputs of daily global T2m air temperature and a daily regional synoptic principal component score. Three of the models use soil moisture 90-day running mean soil moisture—two integrated

throughout the total soil column and one for the surface (0.05m) layer-only. With their chosen ensemble, the authors isolate separately the role of soil moisture and hidden layer count in ANN performance. The documented exercise and discussion serves to elevate the importance of considering the coupled (T and Q) thermodynamic variables in warming and heat wave analyses. The main findings of the paper are that (1) more complex, 3-layer ANN models offer superior performance related to linear (MLR) and less-complex, 1-layer ANN models and (2) soil moisture inputs are critical to predicting equivalent potential maximums, especially in semi-arid regions of strong land-atmosphere coupling or 'warming-holes'. The ANN models presented could be used in a forecast mode to transfer forecasted synoptic patterns (T850, Q850, H500) and soil moisture into near-surface equivalent potential temperature forecasts, but one wonders if the T2, Q2, and Psurf would not be available in the forecast to compute equivalent potential directly, especially if these fields are available at 850hPa. The article is generally well written and I appreciate the perspective on equivalent potential temperature being offered, which is that of the appropriate measure for heat severity and static energy variability. This paper demonstrates the predictability of equivalent potential, however, makes no strong linkages to standard measure of human heat indices.

I am providing below a number of minor issues that should be resolved prior to publication.

Abstract: "...measure of static energy...are more strongly linked to excess human mortality and morbidity than air temperature alone"- while this statement provides sound motivation for the study, the two are only weakly connected in the article's discussion and not at all connected in the results. Connecting this idea back in for the Results and Conclusion would be desirable.

Ln 16 model[s]

Ln17 the "drivers", should be explicitly named in the abstract: global T2m, synoptic T,

Q at 850 and 500 hPa geopotential, and SM—all taken from MERRA2.

Ln 25 "Over the [eastern U.S.], the ANN…alarm rate [is] ~0.08."

Main Body Pg1ln30 "elaborated [on] the drivers...health [and socioeconomic impacts](Sanderson…"

Pg2ln1 "associated with annual [increased] welfare losses of …"- clarify that $57 billion is annual increase due to warming, not annual total.

Pg2ln3 suggest deleting "Many"

Eq1. There should be references to Bolton, 1980; Bryan, 2008; and/or Davies-Jones, 2009. Theta-e should be defined in words here, as well. e.g., "potential temperature plus the temperature increase that would be caused by latent heat of saturation of water contained in air". Suggest starting from potential temperature definition first, then moving to equivalent potential temperature.

Pg2ln14-15 units for each variable need to be added here.

Pg2ln16 should read [g/g], not g/kg in Q

Pg2ln18 isn't T responsive to advection-driven forcing, as well?

Pg2ln22 "[surface broadband] albedo". What is the meaning of "counter-radiated"? Is it "surface downward"?

Pg2ln27 "although [heat extremes] such as…(Garcia-Herrara et al., 2010)…(Vanos et al., 2015)". Other references are required here to strengthen assertion being made. Vanos et al., 2015 covers Midwest only. References list needs to be expanded to cover SE, NE U.S. cases, as well.

Pg2ln31 is Peterson et al., 2011 reference for 2m static energy? Please clarify level to which "lower atmosphere" refers.

Pg3ln2 Davey et al., 2006 reference expected after Eq. 1.

Pg3ln3 statistical models are ill-suited for pursuits of physical, process-level understanding

Pg3ln6 suggest using the more common equivalent term of "summary"

Pg3ln27 "extreme [warm] air T"

Pg3ln34 "extreme [high] Td"

Pg3ln36 "High Plains to [] the upper Great Lakes". Restricted vertical mixing = subsidence?

Pg4lns2-4 scale length for (b) and (c) need also to be specified as it is for (a) (i.e., global).

Pg4ln11 technically, Canada should be masked out of Figure 1a. Domain lat-lon extents should be provided here, as on pg5ln16

Pg4ln12 "trends in Te" please specify over what period these trends were computed.

Pg4ln15 could you provide an estimate of the affected population residing in the eastern U.S. domain?

Pg4ln19 sentence beginning "Therefore,.." is confusing and should be reworded.

Pg4ln21 it is confusing to bundle land management and SM rates of change in the same sentence while they occur on very different time scales. Clarify relative rate of change and period of change.

Pg4ln23 specify the "Parts of the region" that are being referenced. Is this the southern Great Plains? There is no reason to be non-specific here. The GLACE hotspot and two regions of reduced Tmax should be demarcated on Fig 1a, which could be enlarged.

Pg4ln26 Guo et al., 2006 and Dirmeyer and Halder (2017) should be added to reference list.

Pg4ln27 "The study region". The preceding three bullets referred to the study region

as "it". Why the change here? Also regarding: "maximum T during parts or all of the twentieth century", it is unclear whether "parts" in this context alludes to seasons, years, consecutive years, etc. This should be clarified.

Pg4ln29 "lack of warming" The warming hole label was included in the abstract and this is probably the appropriate place to introduce the term in the article.

Pg4ln31 suggest new sentence begins with "...Ellenburg et al., 2016). In the case of Mississippi...(AL), [up to ]..."

Pg4ln32 specify whether "summer temperature" is JJA T, JJA Tmax, JJA Tmin, etc. This section should be more carefully worded and details added to improve clarity of meaning.

Pg5ln7 consider adding a note that MERRA-2 uses bias-corrected P to drive the land surface model, which lends strong confidence to the SM estimates. The skill of MERRA-2 lower-atmospheric fields used in the synoptic airflow classification is yet to be well established.

Pg5ln16 suggest adding 12 EDT /2 CDT after 20 UTC.

Pg5ln30 "strong [southerly] low-level advection of [high] T and Q into the region"?

Pg5ln36 "[Due to its spatial heterogeneity,] soil moisture is ..."

Pg6ln3 SM plays much less of a role in the radiation-limited temperate NE forests.

Pg6ln6 I would not agree that MERRA-2 SM has been "extensively" evaluated- it is a very new product and only one (biased) reference is provided.

Pg6ln8 it should be specified that validation occurs only over non-forested sites, where in-situ SM is available.

Pg6ln18 was the time difference (2PM EDT, 1PM CDT) accounted for in the comparison? If so, how?

Pg6ln19 is the GPLLJ the only source of moisture for MO and IA? Sentence is currently misleading.

Pg6ln21 "and the presence of abundant SM". Intent of statement is unclear. SM is abundant in the eastern states, as well.

Pg6ln21 suggest list item (5) should be added, beginning from "There are also important..."

Pg6ln25-30 suggest moving these stats out of the data section and into Results.

Pg6ln30 "over [the] southern..."

Pg7ln8 please include commentary on how these predictors were selected. What was the logic or metrics employed in the selection process?

Pg7ln24 "readily available to [soil evaporation, as opposed to integrated soil moisture profile that constitutes the water availability to evapotranspiration]".

Pg7ln25 suggest "Table 1 summarizes the acronyms used herein for each of the five models considered. A schematic of the model architecture and data flows is provided in Fig. 3.

Pg7ln27 suggest including statement that the 70-15-15 sample subsets were identical for all model architectures.

Pg8ln17 this statement appears at odds with Table 1, which lists ANN-HL3 as having the worst model statistics.

Pg9ln2 is there any statistical significance to this difference?

Pg9ln6 "where [modeled] land-atmosphere...and where strong [longitudinal] gradients of SM"

Ph9ln13 "such [as] dry lines..."

Pg9ln24 I question "evaporation from the Great Lakes". Is Great Lakes evaporation a

strong predictor of theta-e in the eastern U.S.? Evaporation over the Great Lakes is much more substantial in the winter months and drives tremendous lake-effect snow bands, but the affected region is isolated.

Pg9ln30 how is water management signal being linked to the present analysis? MERRA2 has no accounting of water management.

Pg9ln32 suggest DeAngelis et al. 2010 in addition to Pryor et al., 2016 reference for warming hole

Pg10ln5 HL3-SM seems to perform the best; 3 hidden layers as opposed to 1.

Pg10ln7 "...when all [eastern U.S.] grid cells are considered..."

Pg10ln12 please clarify that "test period" here is synonymous with "independent sample", or 15% of all JJA days

Pg10lns5-36. Why can't these stats on HR and FAR be added to an expanded Table 1 for each model?

Pg10ln22 statement on CLM (Buzan et al., 2015) does not appear directly relevant to MERRA-specific results being discussed here. It is out of place, given non-MERRA results are not specifically called out elsewhere in the paper. I suggest removing this statement.

Pg10ln26 To include MO, IA and IL is a stretch. The central U.S. hotspot of GLACE-1 was squarely contained within 27-41N; 106-97W.

Pg10ln32 please quantify "greatly" Is this statistically significant?

Pg11ln1 "HR and FAR are comparable to (or better than) seasonal re-forecasts of summertime T at 2-m..." Please clarify were the HR and FAR of T or theta-e-max,min compared against the HR and FAR of T2m from ECMWF (ERA-I?)?

Pg11ln15 the comparison between HL3-SM and HL3-TOP needs to be included and

thoroughly discussed, esp. for theta-e-max.

Pg11ln32 "from the [MERRA-2]. . ."

Pg11ln33 suggest "important differences in the magnitude of derived equivalent temperature (Te)[,as well as in strength of land-atmosphere coupling between the reanalysis products (e.g., Ferguson et al., 2012; Schoof et al., 2017)]"

Pg12ln9 sentence beginning "Correlation coefficients exceed. . ." needs to be reworded and probably split into two or more sentences.

Pg12ln11 sentence beginning "This is true for the simulation. . ." is confusing. Does this imply that the prior sentence holds exactly for max theta e?

Pg12ln15 the statement that "min theta-e exhibits a stronger dependence on the precise prevailing synoptic scale conditions" is unsupported by the analysis and accompanying discussion. Where has the link between min theta-e predictability and PC number been established and probed?

Pg12ln28 what are the alternative hypotheses in literature for the 'warming hole'? references here or previously upon the introduction of the term would be useful.

Pg12ln30 I disagree that the statistical modeling exercise documented herein has "enhance[d] mechanistic understanding of the causes of variability and change in theta-e". This claim should be deleted.

Pg12ln35. The authors should comment on lessons learned and insight gleaned. For example, the weighting for the global T was stated to be negligible. Then, is their recommendation to forgo global T requirements in future ANN pursuits? Similarly, is the total integrated column soil moisture necessary or merely the surface layer? Does the fact that MERRA2 uses bias-corrected P make any allusions of similar success in full forecast models (with biased, model P) unfair and misleading? In addition to those predictors tested here, which other predictors would be meaningful to explore/consider?

Table 1. Caption should specify over JJA and 1980-2014; "coefficient [(r)]", "the presence [or] absence"; the total number of grids (1962) should be specified OR the stats should be presented as percentages to be consistent with the text. Column with r>0.8 and RMSE <5k: what about the number of grids that satisfy BOTH criteria?

Fig 1. For (c-g) it is unclear from the caption whether 1980-2014 is still the averaging period; "standard deviation of daily [JJA] (e)..."; Mean [JJA daily] soil moisture...'; explained computation would be more appropriate to include in the main text of article. Is SM also averaged over 1980-2014? ; "total profile [0-1m? profile total thickness]"; the variable names and units should be displayed on each subpanel. Is the 90-day running JJA just June1-Aug31 average or on June 1 the ~March1-June1 average and so-on? Please clarify.

Fig3. The 5 models from Table 1 should be alluded to here by their acronyms in that table; the domain of actions on the right hand side should be specified as for the predictors. From the flowchart is appears that z-scores are fed into the ANN. From the text I understood that daily z-scores are used to compute daily PC's which are passed to the ANN. This Figure could be eliminated by adding a data table, which demarcates the predictors and predictands, and describing the rest in text. Currently the flowchart is a bit unclear.

Fig 4. ANN-HL3-TOP is missing and needs to be included. The labels and units for rows 2 and 3 should be added to the far right side.

References (Bolton 1980; Bryan 2008; Davies-Jones 2009; DeAngelis et al. 2010; Dirmeyer and Halder 2017; Ferguson et al. 2012; Guo et al. 2006)

Bolton, D., 1980: The Computation of Equivalent Potential Temperature. Mon Weather Rev, 108, 1046-1053.

Bryan, G. H., 2008: On the Computation of Pseudoadiabatic Entropy and Equivalent Potential Temperature. Mon Weather Rev, 136, 5239-5245.

Davies-Jones, R., 2009: On Formulas for Equivalent Potential Temperature. Mon Weather Rev, 137, 3137-3148.

DeAngelis, A., F. Dominguez, Y. Fan, A. Robock, M. D. Kustu, and D. Robinson, 2010: Evidence of enhanced precipitation due to irrigation over the Great Plains of the United States. J Geophys Res-Atmos, 115.

Dirmeyer, P. A., and S. Halder, 2017: Application of the Land–Atmosphere Coupling Paradigm to the Operational Coupled Forecast System, Version 2 (CFSv2). J Hydrometeorol, 18, 85-108.

Ferguson, C. R., E. F. Wood, and R. K. Vinukollu, 2012: A Global Intercomparison of Modeled and Observed Land-Atmosphere Coupling. J Hydrometeorol, 13, 749-784.

Guo, Z. C., and Coauthors, 2006: GLACE: The Global Land-Atmosphere Coupling Experiment. Part II: Analysis. J Hydrometeorol, 7, 611-625.

---

## Referee Comment (RC2) · Anonymous Referee #3 · 23 Oct 2017

This study examined different statistical models for summertime daily maximum and minimum $\theta$e to understand the drivers of historical changes and variability over the eastern USA. The non-linear model considering soil moisture show improved $\theta$e estimations over majority of the study region. The authors also argued that SM has played a key role in dictating the presence and intensity of the "warming holes". I think this work will make a valuable contribution to understand the drivers of surface static energy and heat waves, and would recommend publication after some minor revisions. My comments are listed below:

[Figure]

1) There major drivers (e.g., global temperature, synoptic-scale indices and SM) were used to develop statistical models for $\theta$e. By comparing the statistical models with and without considering the SM (e.g., ANN-HL3-SM vs ANN-HL3), the authors concluded that the SM played a key role in dictating the warming holes. This statement may be misleading. To identify the relative roles of individual drivers, it might be better to develop statistical models by examining different combinations of the drivers (e.g., global temperature and SM, or synoptic-scale indices and SM). It is also possible that the roles of different drivers may vary in different regions. 2) The climate variations in the eastern USA are influenced by different climate modes (e.g., ENSO, NAO, IPO). The aerosol may also play some roles on the warming holes. It is not clear why these modes were not used in this study. 3) For SM index. The 90-day running mean estimate of antecedent SM in 3x3 grid cells were used. Why it is necessary use the 90-day running mean and 3x3 grid cells average? It was found that the SM would influence the climate downstream. Do you think it is possible to improve the model results by averaging the SM over a large region (e.g., 5x5 or 10x10 grid cells)? 4) Page 4, line 15. How were the daily maximum and minimum $\theta$e calculated? Did you firstly compute the $\theta$e using the hourly T, q, and P, and then derive the maximum and minimum values? 5) Page 5, last sentence. The global T and 15 synoptic-scale PC scores are common to models built for all grid cells, whereas the SM is grid-cell specific. Therefore, it is not a surprise that the SM plays a more important role in the model performance. 6) Table 1. The # of grid cells with r>0.8 & RMSE < 5K were shown. However, it might be better to show the percentage because the number of grid cells depends on the spatial resolution of the dataset used. 7) Figure 2. Would it possible to add the state or continental boundary to the figures?
* * *

---

## Author Comment (AC1) · 1 Nov 2017

Response to: Interactive comment on "Modeling the contributions of global air temperature, synoptic-scale phenomena and soil moisture to near-surface static energy variability using artificial neural networks" by Sara C. Pryor et al. Anonymous Referee #2

Below we list the comments of the reviewer and our responses (in italics below each point). At the end of this document we also provide a tracked changes version of the

manuscript showing the changes we have made in full. Note: a pdf of this document is uploaded as a 'supplement' to this response.

. . .<Preamble deleted for brevity> . . .

I am providing below a number of minor issues that should be resolved prior to publication.

Abstract: "... measure of static energy... are more strongly linked to excess human mortality and morbidity than air temperature alone"- while this statement provides sound motivation for the study, the two are only weakly connected in the article's discussion and not at all connected in the results. Connecting this idea back in for the Results and Conclusion would be desirable Response: Done. We have extensively modified section 4 to include the requested linkages back to human health.

Ln 16 model[s] Response: Done

Ln17 the "drivers", should be explicitly named in the abstract: global T2m, synoptic T Q at 850 and 500 hPa geopotential, and SMâAËĞT all taken from MERRA2. Response: Done. Added text: The predictor variables include an index of near-surface daily global mean temperature, daily indices of the synoptic scale meteorology derived from T and specific humidity (Q) at 850 hPa and 500 hPa geopotential heights (Z), and spatiotemporally averaged soil moisture (SM).

Ln 25 "Over the [eastern U.S.], the ANN...alarm rate [is] âĹ́ij0.08." Response: Done

Main Body Pg1ln30 "elaborated [on] the drivers...health [and socioeconomic impacts](Sanderson..." Response: Done

Pg2ln1 "associated with annual [increased] welfare losses of ..."- clarify that $57 billion is annual increase due to warming, not annual total. Response: We do not refer to welfare losses of $57 billion, so no clarification needed.

Pg2ln3 suggest deleting "Many" Response: Done

Eq1. There should be references to Bolton, 1980; Bryan, 2008; and/or Davies-Jones, 2009. Theta-e should be defined in words here, as well. e.g., "potential temperature plus the temperature increase that would be caused by latent heat of saturation of water contained in air". Suggest starting from potential temperature definition first, then moving to equivalent potential temperature. Response: Reference added along with following text; 'Potential temperature is the temperature an unsaturated air parcel would have if brought adiabatically to a standard pressure. Thus, potential temperature is conserved for an unsaturated air parcel if it remains unsaturated as it rises and sinks. Equivalent potential temperature is conserved under vertical motion even if there is phase change of water vapor contained within the air. Use of metrics such as $\theta$e permits more accurate depictions of near-surface energy budgets and surface heating trends (Davey et al., 2006)..'

Pg2ln14-15 units for each variable need to be added here. Response: Done

Pg2ln16 should read [g/g], not g/kg in Q Response: Can not find what the reviewer was referring to we did not state units of g/kg, but addressed by the above.

Pg2ln18 isn't T responsive to advection-driven forcing, as well? Response: Yes. We have reworded this to clarify that here we are referring to the local surface energy balance prior to describing non-local processes in the following paragraphs. The rewrite reads; 'When considering the local surface energy balance (Eq. 2), near-surface T is responsive only to changes in the sensible heat flux from/to the surface, while $\theta$e responds to changes in both the sensible and latent heat flux terms.'

Pg2ln22 "[surface broadband] albedo". What is the meaning of "counter-radiated"? Is it "surface downward"? Response: Yes, "counter-radiated longwave radiation from the atmosphere" is 'surface downward' but I believe 'counter-radiated longwave radiation from the atmosphere' is in common usage in atmospheric science and thus should be clear to the readership of ACP.

Pg2ln27 "although [heat extremes] such as ... (Garcia-Herrara et al., 2010)... (Vanos

et al., 2015)". Other references are required here to strengthen assertion being made. Vanos et al., 2015 covers Midwest only. References list needs to be expanded to cover SE, NE U.S. cases, as well. Response: The following paper that we also cite does cover cities drawn from the entire contiguous USA: Anderson, G. B., and Bell, M. L.: Heat waves in the United States: Mortality risk during heat waves and effect modification by heat wave characteristics in 43 U.S. communities, Environmental Health Perspectives, 119, 210-218, 2011.

Pg2ln31 is Peterson et al., 2011 reference for 2m static energy? Please clarify level to which "lower atmosphere" refers. Response: This is an excellent point. Peterson et al. 2011 refer to 'surface atmospheric energy ' (in the title of their article) but do not specify the height they draw T, Q and wind speeds from in order to compute the components of the atmospheric energy budget. I suspect (based on NWS protocols in the USA) that T and Q are LIKELY to be drawn from 2-m, but wind speeds are more likely to be taken at 10-m. However, the reviewer is correct, the lack of specificity on the part of those authors led me to state "lower atmosphere" in the absence of more specific information. I have changed this to near-surface to be consistent with the authors to whose work we are referring.

Pg3ln2 Davey et al., 2006 reference expected after Eq. 1 Response: Done

Pg3ln3 statistical models are ill-suited for pursuits of physical, process-level understanding Response: Respectfully, this is only partly true. Statistical relationships MAY not be reflective of an underlying physical process. HOWEVER, the fundamental basis of all big-data analytics and deep-learning (statistical) data analysis methods is that one can extract high-level, complex abstractions as data representations through a hierarchical learning process. This implies there are repeatable associations that are manifesting causal mechanisms. Nevertheless, we have removed the word 'mechanistic' from 'enhance mechanistic understanding'.

Pg3ln6 suggest using the more common equivalent term of "summary" Response: I

can not find what the reviewer was referring to ... Pg3 line 6 read; 'the summer of 2012 in part due to persistent anticyclonic conditions (Peterson et al., 2013). Further, variability'

Pg3ln34 "extreme [high] Td" Response: Done

Pg3ln36 "High Plains to [] the upper Great Lakes". Restricted vertical mixing = subsidence? Response: This is an interesting point. The authors of the article to which we are referring actually DID NOT evaluate subsidence but do report 'restricted vertical mixing'. Thus we have maintained their statement (restricted vertical mixing) and not inferred information about an increase in subsidence.

Pg4lns2-4 scale length for (b) and (c) need also to be specified as it is for (a) (i.e., global). Response: Done

Pg4ln11 technically, Canada should be masked out of Figure 1a. Domain lat-lon extents should be provided here, as on pg5ln16 Response: Although our primary focus is on the eastern USA and perhaps we should mask out Canada, we prefer to include grid cells within Canada since there have been heat-wave incidents in Canada (see (Smoyer-Tomic et al., 2003)). I have added the domain extent to the caption to Figure 1.

Pg4ln12 "trends in Te" please specify over what period these trends were computed. Response: Done (its 1981-2015).

Pg4ln15 could you provide an estimate of the affected population residing in the eastern U.S. domain? Response: the eastern US (as defined in our study) has a population of > 200 million (we have added this to the text; 'It is home to over 200 million people (based on the 2013 census).'

Pg4ln19 sentence beginning "Therefore,.." is confusing and should be reworded. Response: Unfortunately there is no sentence on page 4 line 19 that begins 'therefore'.

Pg4ln21 it is confusing to bundle land management and SM rates of change in the

same sentence while they occur on very different time scales. Clarify relative rate of change and period of change. Response: We believe the reviewer is referring to 'It exhibits strong spatial gradients in terms of the nature of land cover and rate of change of both land management and soil moisture.' They are quite correct, these are complex processes. We have added the following: 'For example, over the period 1950-2000 the region as a whole experienced rapid population growth (though this was not spatially uniform), expansion of area classified as exurban, an overall reduction of land in agriculture (though again this was highly heterogeneous in space) and an increase in the intensity of water management (including expansion of irrigation) (Brown et al., 2005).'

Pg4ln23 specify the "Parts of the region" that are being referenced. Is this the southern Great Plains? There is no reason to be non-specific here. The GLACE hotspot and two regions of reduced Tmax should be demarcated on Fig 1a, which could be enlarged. Response: At least to our knowledge the two references we cite not provide precise lat/long coordinates for the regions of strong coupling (though they are shown on maps). We note the reviewer did state (later in their review); 'The central U.S. hotspot of GLACE-1 was squarely contained within 27-41N; 106-97W'... In order to address this concern we have elaborated in the text so that it now reads: "Parts of the region (focused on the southern Great Plains, i.e. the southern and western-most portion of the current study area) were identified in the global land–atmosphere coupling experiments (GLACE) as exhibiting atypically strong atmosphere-surface coupling in some global climate models. Further, soil moisture makes a large contribution to sub-seasonal forecast skill for air temperatures and precipitation in this region (Koster et al., 2011;Koster et al., 2006)."

Pg4ln26 Guo et al., 2006 and Dirmeyer and Halder (2017) should be added to reference list. Response: Done

Pg4ln27 "The study region". The preceding three bullets referred to the study region as "it". Why the change here? Also regarding: "maximum T during parts or all of the

twentieth century", it is unclear whether "parts" in this context alludes to seasons, years, consecutive years, etc. This should be clarified. Response: Each bullet now started with it. We have clarified; 'It also incorporates two areas of reduced daily maximum T during multiple consecutive years extending over part or all of the twentieth century.'

Pg4ln29 "lack of warming" The warming hole label was included in the abstract and this is probably the appropriate place to introduce the term in the article. Response: Done

Pg4ln31 suggest new sentence begins with "... Ellenburg et al., 2016). In the case of Mississippi ... (AL), [up to ]..." Response: Done

Pg4ln32 specify whether "summer temperature" is JJA T, JJA Tmax, JJA Tmin, etc. This section should be more carefully worded and details added to improve clarity of meaning. Response: Where hourly data are used it is specified (e.g. for T, Q and P).

Pg5ln7 consider adding a note that MERRA-2 uses bias-corrected P to drive the land surface model, which lends strong confidence to the SM estimates. The skill of MERRA-2 lower-atmospheric fields used in the synoptic airflow classification is yet to be well established. Response: Yes, excellent suggestion. Done. In terms of the evaluation of MERRA-2 – this is clearly NOT our product but NASA scientists have been extensively evaluating the reanalysis system and output. What we can assert is that the output from our PCA analysis of the synoptic scale meteorology generates reasonable and interpretable types. This is why we wrote (and write); 'As shown, many of the synoptic types thus identified are readily interpretable as representing a diversity of zonal versus meridional circulation (cf. type 9 and 5 and 6), and some are characterized by conditions known to be associated with strong low-level advection of T and q into the region (e.g. types 1, 3, and 15) (Pryor and Schoof, 2016;Weaver, 2013). Further, most types exhibit a high degree of similarity with other synoptic-scale classifications derived for the region (e.g. type 7 is very similar to one that is associated with summertime precipitation over the southeastern USA (Diem, 2006)).' Naturally,

this does not constitute a comprehensive evaluation!

Pg5ln16 suggest adding 12 EDT /2 CDT after 20 UTC. Response: We have added a note regarding time zones. Thus the text now read; 'for 20:00 UTC (i.e. 16:00 Eastern Daylight Time, 15:00 Central Daylight Time)'. Please note: Eastern Daylight Time (EDT) is UTC MINUS 4 hr, not 8 hours as the reviewer wrote (maybe this is a typographic error).

Pg5ln30 "strong [southerly] low-level advection of [high] T and Q into the region"? Response: Done

Pg5ln36 "[Due to its spatial heterogeneity,] soil moisture is... " Response: Done

Pg6ln3 SM plays much less of a role in the radiation-limited temperate NE forests. Response: Yes.

Pg6ln6 I would not agree that MERRA-2 SM has been "extensively" evaluated- it is a very new product and only one (biased) reference is provided Response: We have removed 'extensively'.

Pg6ln8 it should be specified that validation occurs only over non-forested sites, where in-situ SM is available. Response: Very good point – well made. We have added a caveat that reads; 'However, it should be noted that there are relatively few direct measurements of SM and thus the evaluation of MERRA-2 is focused on agricultural locations.'

Pg6ln18 was the time difference (2PM EDT, 1PM CDT) accounted for in the comparison? If so, how? Response: No.

Pg6ln19 is the GPLLJ the only source of moisture for MO and IA? Sentence is currently misleading. Response: Fair point we have add 'in part' to clarify.

Pg6ln21 "and the presence of abundant SM". Intent of statement is unclear. SM is abundant in the eastern states, as well. Response: This is a very interesting point.

There are relatively few homogeneous SM data sets. IF (and I would emphasize, if) one treats remote sensing SM estimates as equally valid over the entire study area then for our study region SM is most abundant in the region I was referring to (lower GP) and in the NE, but is less abundant in the eastern states. See figure below:

Image from: https://www.jpl.nasa.gov/spaceimages/details.php?id=PIA19337. Caption: High-resolution global soil moisture map from SMAP's combined radar and radiometer instruments, acquired between May 4 and May 11, 2015 during SMAP's commissioning phase. The map has a resolution of 5.6 miles (9 kilometers). We have added a reference to (Al Bitar et al., 2017) in support of the assertion regarding abundant SM so we are not solely relying on the MERRA-2 product (and our figure 1).

Pg6ln21 suggest list item (5) should be added, beginning from "There are also Important ..." Response: Done

Pg6ln25-30 suggest moving these stats out of the data section and into Results. Response: We decided against doing so.

Pg6ln30 "over [the] southern ..." Response: Done

Pg7ln8 please include commentary on how these predictors were selected. What was the logic or metrics employed in the selection process? Response: Done (but we do so where the predictors are first introduced – i.e. in the bulleted list in section 2.2 not where the reviewer suggested).

Pg7ln24 "readily available to [soil evaporation, as opposed to integrated soil moisture profile that constitutes the water availability to evapotranspiration]". Response: We have modified the sentence to read; 'Therefore, in addition to developing models using the MERRA-2 variable 'PRMC', which is the 'Total profile soil moisture content' in m3m-3 (that is summed across all six soil layers and represents the total water potentially available for evapotranspiration to the atmosphere), a fifth ANN model (with 3 hidden layers) is also built that uses the variable 'GWETTOP' that describes the SM content

in the upper 5 cm of the soil (unitless) (Reichle et al., 2017), and thus best represents the SM that is readily available for evaporation into the overlying atmosphere.'

Pg7ln25 suggest "Table 1 summarizes the acronyms used herein for each of the five models considered. A schematic of the model architecture and data flows is provided in Fig. 3. Response: Done

Pg7ln27 suggest including statement that the 70-15-15 sample subsets were identical for all model architectures. Response: Done

Pg8ln17 this statement appears at odds with Table 1, which lists ANN-HL3 as having the worst model statistics. Response: I believe this is a misunderstanding ANN-HL3 excludes SM and indeed has highest RMSE.

Pg9ln2 is there any statistical significance to this difference? Response: The reviewer is raising an interesting point – can one undertake an appropriate non-parametric test, correcting (of course) for multiplicity. We did not, instead choosing to draw out spatial consistency as a basis for inferring 'skill'.

Pg9ln6 "where [modeled] land-atmosphere ... and where strong [longitudinal] gradients of SM" Response: Done

Ph9ln13 "such [as] dry lines ..." Response: Done

Pg9ln24 I question "evaporation from the Great Lakes". Is Great Lakes evaporation a strong predictor of theta-e in the eastern U.S.? Evaporation over the Great Lakes is much more substantial in the winter months and drives tremendous lake-effect snow bands, but the affected region is isolated. Response: Interesting point. As a resident of upstate New York I can confirm the lake-effect snows are a strong function of ice cover (and hence evaporation) from the Great Lakes! However, observationally-derived estimates of evaporation from the Great Lakes reported in (Notaro et al., 2015) are certainly 'non-trivial' (i.e. approx. an average of 45 mm in August, relative to over-lake precipitation receipt of approx. 85 mm for the period 1980-1999..). But it is an open

question as to the impact on downstream theta-e, so we have added 'potentially'.

Pg9ln30 how is water management signal being linked to the present analysis? MERRA2 has no accounting of water management. Response: Right, its not.

Pg9ln32 suggest DeAngelis et al. 2010 in addition to Pryor et al., 2016 reference for warming hole Response: Done

Pg10ln5 HL3-SM seems to perform the best; 3 hidden layers as opposed to 1. Response: Correct

Pg10ln7 "...when all [eastern U.S.] grid cells are considered..." Response: Done

Pg10ln12 please clarify that "test period" here is synonymous with "independent sample", or 15% of all JJA days Response: Reworded to independent data sample.

Pg10lns5-36. Why can't these stats on HR and FAR be added to an expanded Table 1 for each model? Response: This is a style consideration. Table 1 is designed to summarize the overall performance of the models (and is already quite complex), the HR/FAR naturally reflect the performance for extremes.

Pg10ln22 statement on CLM (Buzan et al., 2015) does not appear directly relevant to MERRA-specific results being discussed here. It is out of place, given non-MERRA results are not specifically called out elsewhere in the paper. I suggest removing this statement. Response: It is POTENTIALLY interesting at least to some readers to note that; 'The causes of the poor model performance in eastern TX and SC are currently not fully understood, although it is worthy of note that data from MERRA-2 grid-cells in SC exhibit a relatively low overall frequency of exceedance of this threshold and are also characterized by comparatively low 99th percentile $\theta$e in an analysis of heat indices derived from the Community Land Model v4.5 (Buzan et al., 2015).'

Pg10ln26 To include MO, IA and IL is a stretch. The central U.S. hotspot of GLACE-1 was squarely contained within 27-41N; 106-97W. Response: Right, the other references we cite do focus more on the eastern states. We have reworded to; 'central and

eastern USA'

Pg10ln32 please quantify "greatly" Is this statistically significant? Response: Fair point 'greatly' is subjective. I have deleted this word.

Pg11ln1 "HR and FAR are comparable to (or better than) seasonal re-forecasts of summertime T at 2-m..." Please clarify were the HR and FAR of T or theta-e-max,min compared against the HR and FAR of T2m from ECMWF (ERA-I?)? Response: Yes. I have reworded to; 'Further, HR and FAR computed for max-$\theta$e and min-$\theta$e are comparable to (or better than) seasonal re-forecasts of summertime T at 2-m over the land areas of Southern Europe developed using the European Centre for Medium-Range Weather Forecasts (ECMWF) seasonal ensemble forecasting system (Weisheimer et al., 2011).'

Pg11ln15 the comparison between HL3-SM and HL3-TOP needs to be included and thoroughly discussed, esp. for theta-e-max. Response: There is a discussion which reads;' Differences in model performance between ANN conditioned on total SM and using wetness only in the top soil layer (upper 5 cm) are very small when averaged across the domain (Table 1) and indeed for virtually all grid cells. Only 26 grid cells exhibited a $\Delta$|RMSE| > 0.5 K for models using PRMC versus those using GWETTOP (out of a total of 1962), while 155 exhibited an increase in RMSE > 0.5 K when SM was excluded from the model. Thus, although the weights within the ANNs differ for use of the two SM parameters, the overall model skill is unchanged by use of the two SM estimates possibly due to the spatial and temporal averaging applied herein, or uncertainty in reanalysis-derived SM variables.'

Pg11ln32 "from the [MERRA-2]..." Response: Done

Pg11ln33 suggest "important differences in the magnitude of derived equivalent temperature (Te)[,as well as in strength of land-atmosphere coupling between the reanalysis products (e.g., Ferguson et al., 2012; Schoof et al., 2017)]" Response: Done

[Figure]

Pg12ln9 sentence beginning "Correlation coefficients exceed..." needs to be reworded and probably split into two or more sentences. Response: Done

Pg12ln11 sentence beginning "This is true for the simulation..." is confusing. Does this imply that the prior sentence holds exactly for max theta e? Response: Reworded for clarity to ; 'ANN-HL3-SM models also exhibit highest skill for simulation of extreme min- and max-$\theta$e.'

Pg12ln15 the statement that "min theta-e exhibits a stronger dependence on the precise prevailing synoptic scale conditions" is unsupported by the analysis and accompanying discussion. Where has the link between min theta-e predictability and PC number been established and probed? Response: Very good point. We have re-written this sentence to read; 'Results from the ANN models further indicate that max-$\theta$e and the occurrence of extreme max-$\theta$e appear to be considerably more sensitive to SM than min-$\theta$e which in turn appears to exhibit a stronger dependence on the precise prevailing synoptic scale conditions based on the ANN weights.' To reflect the assessment I made of the weights.

Pg12ln28 what are the alternative hypotheses in literature for the 'warming hole'? references here or previously upon the introduction of the term would be useful. Response: Done

Pg12ln30 I disagree that the statistical modeling exercise documented herein has "enhance[d] mechanistic understanding of the causes of variability and change in theta-e". This claim should be deleted. Response: Deleted mechanistic.

Pg12ln35. The authors should comment on lessons learned and insight gleaned. For example, the weighting for the global T was stated to be negligible. Then, is their recommendation to forgo global T requirements in future ANN pursuits? Similarly, is the total integrated column soil moisture necessary or merely the surface layer? Does the fact that MERRA2 uses bias-corrected P make any allusions of similar success in full forecast models (with biased, model P) unfair and misleading? In addition to those pre-

dictors tested here, which other predictors would be meaningful to explore/consider? Response: Well what we actually state is; 'However, it is worthy of note that the weights on the index of global temperatures (and thus expression of internal climate modes) in ANN-HL3-SM are uniformly close to zero across the grid-cells in the interior of the continent, but are of large magnitude in land grid cells close to the Gulf of Mexico (i.e. the south of the domain) and around the Great Lakes.' So no we would not suggest foregoing global T. We have elaborated on 'next steps' and the implications of our work by adding text to the conclusions. Section 4 now read; 
[revised manuscript text omitted]

Table 1. Caption should specify over JJA and 1980-2014; "coefficient [(r)]", "the presence [or] absence"; the total number of grids (1962) should be specified OR the stats should be presented as percentages to be consistent with the text. Column with r>0.8 and RMSE <5k: what about the number of grids that satisfy BOTH criteria? Response: Done and done (new data added to Table 1, and a note added to the text that reads; 'The ANN-HL3-SM model also exhibits the highest number of grid cells that have both a RMSE < 5 K and a r > 0.8 for both max-$\theta$e and min-$\theta$e. (Table 1).').

Fig 1. For (c-g) it is unclear from the caption whether 1980-2014 is still the averaging period; "standard deviation of daily [JJA] (e)..."; Mean [JJA daily] soil moisture..."; explained computation would be more appropriate to include in the main text of article. Is SM also averaged over 1980-2014? ; "total profile [0-1m? profile total thickness]"; the variable names and units should be displayed on each subpanel. Is the 90-day running JJA just June1-Aug31 average or on June 1 the âĹijMarch1-June1 average and so-on? Please clarify. Response: Clarifications added. In the text we have added the following details about SM; 'Specifically we use a 90-day running mean estimate of antecedent SM (i.e. the value for 1 June is an average of values from approx. 1 March to 1 June) over $3\times3$ grid cells centered on the grid cell in question (i.e. integrated over an area of

approximately 30,000 km2, see Figure 1g).' This is already a relatively compact figure so have maintained the units in the caption.

Fig3. The 5 models from Table 1 should be alluded to here by their acronyms in that table; the domain of actions on the right hand side should be specified as for the predictors. From the flowchart is appears that z-scores are fed into the ANN. From the text I understood that daily z-scores are used to compute daily PC's which are passed to the ANN. This Figure could be eliminated by adding a data table, which demarcates the predictors and predictands, and describing the rest in text. Currently the flowchart is a bit unclear. Response: I've remade the figure to define the acronyms and clarify the data flows.

Fig 4. ANN-HL3-TOP is missing and needs to be included. The labels and units for rows 2 and 3 should be added to the far right side. Response: ANN-HL3-TOP is qualitatively so similar to ANN-HL3-SM that in order to enhance the legibility of the figure I excluded it. I have added a note to the caption to reflect this. This is already a relatively compact figure so have maintained the units in the caption.

References (Bolton 1980; Bryan 2008; Davies-Jones 2009; DeAngelis et al. 2010; Dirmeyer and Halder 2017; Ferguson et al. 2012; Guo et al. 2006) . . .<References supplied by the reviewer deleted for brevity> . . .

References cited in this response Al Bitar, A., Mialon, A., Kerr, Y. H., Cabot, F., Richaume, P., Jacquette, E., Quesney, A., Mahmoodi, A., Tarot, S., Parrens, M., Al-Yaari, A., Pellarin, T., Rodriguez-Fernandez, N., and Wigneron, J. P.: The global SMOS Level 3 daily soil moisture and brightness temperature maps, Earth System Science Data, 9, 293-315, 10.5194/essd-9-293-2017, 2017. Brown, D. G., Johnson, K. M., Loveland, T. R., and Theobald, D. M.: Rural land-use trends in the conterminous United States, 1950-2000, Ecological Applications, 15, 1851-1863, 10.1890/03-5220, 2005. Buzan, J. R., Oleson, K., and Huber, M.: Implementation and comparison of a suite of heat stress metrics within the Community Land Model version 4.5, Geoscientific Model Development, 8, 151-170, 10.5194/gmd-8-151-2015, 2015. Davey, C. A., Pielke, R. A., Sr., and Gallo, K. P.: Differences between near-surface equivalent temperature and temperature trends for the Eastern United States - Equivalent temperature as an alternative measure of heat content, Global and Planetary Change, 54, 19-32, 10.1016/j.gloplacha.2005.11.002, 2006. Diem, J. E.: Synoptic-scale controls of summer precipitation in the Southeastern United States, Journal of Climate, 19, 613-621, 10.1175/jcli3645.1, 2006. Ferguson, C. R., Wood, E. F., and Vinukollu, R. K.: A Global Intercomparison of Modeled and Observed Land-Atmosphere Coupling, Journal of Hydrometeorology, 13, 749-784, 10.1175/jhm-d-11-0119.1, 2012. Ford, T. W., and Schoof, J. T.: Oppressive Heat Events in Illinois Related to Antecedent Wet Soils, Journal of Hydrometeorology, 17, 2713-2726, 2016. Ford, T. W., and Schoof, J. T.: Characterizing extreme and oppressive heat waves in Illinois, Journal of Geophysical Research: Atmospheres, 122, 682-698, 2017. Koster, R. D., Sud, Y., Guo, Z., Dirmeyer, P. A., Bonan, G., Oleson, K. W., Chan, E., Verseghy, D., Cox, P., and Davies, H.: GLACE: the global land–atmosphere coupling experiment. Part I: overview, Journal of Hydrometeorology, 7, 590-610, 2006. Koster, R. D., Mahanama, S., Yamada, T., Balsamo, G., Berg, A., Boisserie, M., Dirmeyer, P., Doblas-Reyes, F., Drewitt, G., and Gordon, C.: The second phase of the global land–atmosphere coupling experiment: soil moisture contributions to subseasonal forecast skill, Journal of Hydrometeorology, 12, 805-822, 2011. McKinnon, K. A., Rhines, A., Tingley, M. P., and Huybers, P.: Long-lead predictions of eastern United States hot days from Pacific sea surface temperatures, Nature Geoscience, 9, 389-394, 10.1038/ngeo2687, 2016. Meehl, G. A., Arblaster, J. M., and Chung, C. T. Y.: Disappearance of the southeast US "warming hole" with the late 1990s transition of the Interdecadal Pacific Oscillation, Geophysical Research Letters, 42, 5564-5570, 10.1002/2015gl064586, 2015. Notaro, M., Bennington, V., and Lofgren, B.: Dynamical Downscaling-Based Projections of Great Lakes Water Levels, Journal of Climate, 28, 9721-9745, 10.1175/jcli-d-14-00847.1, 2015. Peterson, T. C., Alexander, L. V., Allen, M. R., Anel, J. A., Barriopedro, D., Black, M. T., Carey-Smith, T., Castillo, R., Cattiaux, J., Chen, X. L., Chen, X. Y.,

Chevallier, M., Christidis, N., Ciavarella, A., de Vries, H., Dean, S. M., Deans, K., Diffenbaugh, N. S., Doblas-Reyes, F., Donat, M. G., Dong, B., Eilerts, G., Funk, C., Galu, G., Garcia-Herrera, R., Germe, A., Gill, S., Gimeno, L., Guemas, V., Herring, S. C., Hoell, A., Hoerling, M. P., Huntingford, C., Husak, G., Imada, Y., Ishii, M., Karoly, D. J., Kimoto, M., King, A. D., Knutson, T. R., Lewis, S. C., Lin, R. P., Lyon, B., Massey, N., Mazza, E., Michaelsen, J., Mollard, J., Mori, M., Mote, P. W., Nieto, R., Otto, F. E. L., Park, J., Perkins, S. E., Rosier, S., Rowland, J., Rupp, D. E., Salas y Melia, D., Scherer, M., Shiogama, H., Shukla, S., Song, F. F., Sparrow, S., Scott, P. A., Sutton, R., Sweet, W., Tett, S. F. B., Trigo, R. M., van Oldenborgh, G. J., van Westrhenen, R., Verdin, J., Watanabe, M., Wittenberg, A. T., Woollings, T., Yiou, P., Zeng, F. R., Zervas, C., Zhang, R., and Zhou, T. J.: Explaining extreme events of 2012 from a climatic perspective, Bulletin of the American Meteorological Society, 94, S1-S74, 10.1175/bams-d-13-00085.1, 2013. Pryor, S. C., and Schoof, J. T.: Evaluation of near‐surface temperature, humidity, and equivalent temperature from regional climate models applied in type II downscaling, Journal of Geophysical Research: Atmospheres, 121, 3326-3338, 2016. Pryor, S. C., Sullivan, R. C., and Wright, T.: Quantifying the roles of changing albedo, emissivity, and energy partitioning in the impact of irrigation on atmospheric heat content, Journal of Applied Meteorology and Climatology, 55, 1699-1706, 2016. Reichle, R. H., Draper, C. S., Liu, Q., Griotto, M., Mahanama, S. P. P., Hoster, R. D., and Lannoy, G. J. M.: Assessment of MERRA-2 land surface hydrology estimates, Journal of Climate, 30, 2937-2960, 2017. Schoof, J. T., Ford, T., and Pryor, S. C.: Recent changes in United States heat wave characteristics derived from multiple reanalyses, Journal of Applied Meteorology and Climatology, In review, 2017. Sheridan, S. C., and Kalkstein, L. S.: Progress in heat watch–warning system technology, Bulletin of the American Meteorological Society, 85, 1931-1941, 2004. Smoyer-Tomic, K. E., Kuhn, R., and Hudson, A.: Heat wave hazards: an overview of heat wave impacts in Canada, Natural hazards, 28, 465-486, 2003. Weaver, S. J.: Factors associated with decadal variability in Great Plains summertime surface temperatures, Journal of Climate, 26, 343-350, 10.1175/jcli-d-11-00713.1,

2013. Weisheimer, A., Doblas‐Reyes, F. J., Jung, T., and Palmer, T.: On the predictability of the extreme summer 2003 over Europe, Geophysical Research Letters, 38, doi: 10.1029/2010GL046455, 2011.

Please also note the supplement to this comment:
https://www.atmos-chem-phys-discuss.net/acp-2017-367/acp-2017-367-AC1-supplement.pdf

[Figure]

**Supplement:**

Below we list the comments of the reviewer and our responses (in italics below each point). At the end of this document we also provide a tracked changes version of the manuscript showing the changes we have made in full.

…*<Preamble deleted for brevity>* …

I am providing below a number of minor issues that should be resolved prior to publication.

Abstract: "... measure of static energy... are more strongly linked to excess human mortality and morbidity than air temperature alone"- while this statement provides sound motivation for the study, the two are only weakly connected in the article's discussion and not at all connected in the results. Connecting this idea back in for the Results and Conclusion would be desirable
*Response: Done. We have extensively modified section 4 to include the requested linkages back to human health.*

Ln 16 model[s]
*Response: Done*

Ln17 the "drivers", should be explicitly named in the abstract: global T2m, synoptic T Q at 850 and 500 hPa geopotential, and SMâAˇT all taken from MERRA2.
*Response: Done. Added text: The predictor variables include an index of near-surface daily global mean temperature, daily indices of the synoptic scale meteorology derived from T and specific humidity (Q) at 850 hPa and 500 hPa geopotential heights (Z), and spatiotemporally averaged soil moisture (SM).*

Ln 25 "Over the [eastern U.S.], the ANN...alarm rate [is] ~0.08."
*Response: Done*

Main Body Pg1ln30 "elaborated [on] the drivers...health [and socioeconomic impacts](Sanderson..."
*Response: Done*

Pg2ln1 "associated with annual [increased] welfare losses of ..."- clarify that $57 billion is annual increase due to warming, not annual total.
*Response: We do not refer to welfare losses of $57 billion, so no clarification needed.*

Pg2ln3 suggest deleting "Many"
*Response: Done*

Eq1. There should be references to Bolton, 1980; Bryan, 2008; and/or Davies-Jones, 2009. Theta-e should be defined in words here, as well. e.g., "potential temperature plus the temperature increase that would be caused by latent heat of saturation of water contained in air". Suggest starting from potential temperature definition first, then moving to equivalent potential temperature.

*Response: Reference added along with following text; 'Potential temperature is the temperature an unsaturated air parcel would have if brought adiabatically to a standard pressure. Thus, potential temperature is conserved for an unsaturated air parcel if it remains unsaturated as it rises and sinks. Equivalent potential temperature is conserved under vertical motion even if there is phase change of water vapor contained within the air. Use of metrics such as $\theta_e$ permits more accurate depictions of near-surface energy budgets and surface heating trends (Davey et al., 2006)..'*

Pg2ln14-15 units for each variable need to be added here.
*Response: Done*

Pg2ln16 should read [g/g], not g/kg in Q
*Response: Can not find what the reviewer was referring to we did not state units of g/kg, but addressed by the above.*

Pg2ln18 isn't T responsive to advection-driven forcing, as well?
*Response: Yes. We have reworded this to clarify that here we are referring to the local surface energy balance prior to describing non-local processes in the following paragraphs. The rewrite reads; 'When considering the local surface energy balance (Eq. 2), near-surface T is responsive only to changes in the sensible heat flux from/to the surface, while $\theta_e$ responds to changes in both the sensible and latent heat flux terms.'*

Pg2ln22 "[surface broadband] albedo". What is the meaning of "counter-radiated"? Is it "surface downward"?
*Response: Yes, "counter-radiated longwave radiation from the atmosphere" is 'surface downward' but I believe 'counter-radiated longwave radiation from the atmosphere' is in common usage in atmospheric science and thus should be clear to the readership of ACP.*

Pg2ln27 "although [heat extremes] such as ... (Garcia-Herrara et al., 2010)... (Vanos et al., 2015)". Other references are required here to strengthen assertion being made. Vanos et al., 2015 covers Midwest only. References list needs to be expanded to cover SE, NE U.S. cases, as well.
*Response: The following paper that we also cite does cover cities drawn from the entire contiguous USA: Anderson, G. B., and Bell, M. L.: Heat waves in the United States: Mortality risk during heat waves and effect modification by heat wave characteristics in 43 U.S. communities, Environmental Health Perspectives, 119, 210-218, 2011.*

Pg2ln31 is Peterson et al., 2011 reference for 2m static energy? Please clarify level to which "lower atmosphere" refers.

*Response: This is an excellent point. Peterson et al. 2011 refer to 'surface atmospheric energy ' (in the title of their article) but do not specify the height they draw T, Q and wind speeds from in order to compute the components of the atmospheric energy budget. I suspect (based on NWS protocols in the USA) that T and Q are LIKELY to be drawn from 2-m, but wind speeds are more likely to be taken at 10-m. However, the reviewer is correct, the lack of specificity on the part of those authors led me to state "lower atmosphere" in the absence of more specific information. I have changed this to near-surface to be consistent with the authors to whose work we are referring.*

Pg3ln2 Davey et al., 2006 reference expected after Eq. 1
*Response: Done*

Pg3ln3 statistical models are ill-suited for pursuits of physical, process-level understanding
*Response: Respectfully, this is only partly true. Statistical relationships MAY not be reflective of an underlying physical process. HOWEVER, the fundamental basis of all big-data analytics and deep-learning (statistical) data analysis methods is that one can extract high-level, complex abstractions as data representations through a hierarchical learning process. This implies there are repeatable associations that are manifesting causal mechanisms. Nevertheless, we have removed the word 'mechanistic' from 'enhance mechanistic understanding'.*

Pg3ln6 suggest using the more common equivalent term of "summary"
*Response: I can not find what the reviewer was referring to … Pg3 line 6 read; 'the summer of 2012 in part due to persistent anticyclonic conditions (Peterson et al., 2013). Further, variability'*

Pg3ln34 "extreme [high] Td"
*Response: Done*

Pg3ln36 "High Plains to [] the upper Great Lakes". Restricted vertical mixing = subsidence?
*Response: This is an interesting point. The authors of the article to which we are referring actually DID NOT evaluate subsidence but do report 'restricted vertical mixing'. Thus we have maintained their statement (restricted vertical mixing) and not inferred information about an increase in subsidence.*

Pg4lns2-4 scale length for (b) and (c) need also to be specified as it is for (a) (i.e., global).
*Response: Done*

Pg4ln11 technically, Canada should be masked out of Figure 1a. Domain lat-lon extents should be provided here, as on pg5ln16
*Response: Although our primary focus is on the eastern USA and perhaps we should mask out Canada, we prefer to include grid cells within Canada since there have been*

*heat-wave incidents in Canada (see (Smoyer-Tomic et al., 2003)). I have added the domain extent to the caption to Figure 1.*

Pg4ln12 "trends in Te" please specify over what period these trends were computed.
*Response: Done (its 1981-2015).*

Pg4ln15 could you provide an estimate of the affected population residing in the eastern U.S. domain?
*Response: the eastern US (as defined in our study) has a population of > 200 million (we have added this to the text; 'It is home to over 200 million people (based on the 2013 census).'*

Pg4ln19 sentence beginning "Therefore,.." is confusing and should be reworded.
*Response: Unfortunately there is no sentence on page 4 line 19 that begins 'therefore'.*

Pg4ln21 it is confusing to bundle land management and SM rates of change in the same sentence while they occur on very different time scales. Clarify relative rate of change and period of change.
*Response: We believe the reviewer is referring to 'It exhibits strong spatial gradients in terms of the nature of land cover and rate of change of both land management and soil moisture.' They are quite correct, these are complex processes. We have added the following: 'For example, over the period 1950-2000 the region as a whole experienced rapid population growth (though this was not spatially uniform), expansion of area classified as exurban, an overall reduction of land in agriculture (though again this was highly heterogeneous in space) and an increase in the intensity of water management (including expansion of irrigation) (Brown et al., 2005).'*

Pg4ln23 specify the "Parts of the region" that are being referenced. Is this the southern Great Plains? There is no reason to be non-specific here. The GLACE hotspot and two regions of reduced Tmax should be demarcated on Fig 1a, which could be enlarged.
*Response: At least to our knowledge the two references we cite not provide precise lat/long coordinates for the regions of strong coupling (though they are shown on maps). We note the reviewer did state (later in their review); 'The central U.S. hotspot of GLACE-1 was squarely contained within 27-41N; 106-97W'… In order to address this concern we have elaborated in the text so that it now reads: "Parts of the region (focused on the southern Great Plains, i.e. the southern and western-most portion of the current study area) were identified in the global land–atmosphere coupling experiments (GLACE) as exhibiting atypically strong atmosphere-surface coupling in some global climate models. Further, soil moisture makes a large contribution to sub-seasonal forecast skill for air temperatures and precipitation in this region (Koster et al., 2011;Koster et al., 2006)."*

Pg4ln26 Guo et al., 2006 and Dirmeyer and Halder (2017) should be added to reference list.
*Response: Done*

Pg4ln27 "The study region". The preceding three bullets referred to the study region as "it". Why the change here? Also regarding: "maximum T during parts or all of the twentieth century", it is unclear whether "parts" in this context alludes to seasons, years, consecutive years, etc. This should be clarified.
*Response: Each bullet now started with it. We have clarified; 'It also incorporates two areas of reduced daily maximum T during multiple consecutive years extending over part or all of the twentieth century.'*

Pg4ln29 "lack of warming" The warming hole label was included in the abstract and this is probably the appropriate place to introduce the term in the article.
*Response: Done*

Pg4ln31 suggest new sentence begins with "... Ellenburg et al., 2016). In the case of Mississippi ... (AL), [up to ]..."
*Response: Done*

Pg4ln32 specify whether "summer temperature" is JJA T, JJA Tmax, JJA Tmin, etc. This section should be more carefully worded and details added to improve clarity of meaning.
*Response: Where hourly data are used it is specified (e.g. for T, Q and P).*

Pg5ln7 consider adding a note that MERRA-2 uses bias-corrected P to drive the land surface model, which lends strong confidence to the SM estimates. The skill of MERRA-2 lower-atmospheric fields used in the synoptic airflow classification is yet to be well established.
*Response: Yes, excellent suggestion. Done. In terms of the evaluation of MERRA-2 – this is clearly NOT our product but NASA scientists have been extensively evaluating the reanalysis system and output. What we can assert is that the output from our PCA analysis of the synoptic scale meteorology generates reasonable and interpretable types. This is why we wrote (and write); 'As shown, many of the synoptic types thus identified are readily interpretable as representing a diversity of zonal versus meridional circulation (cf. type 9 and 5 and 6), and some are characterized by conditions known to be associated with strong low-level advection of T and q into the region (e.g. types 1, 3, and 15) (Pryor and Schoof, 2016;Weaver, 2013). Further, most types exhibit a high degree of similarity with other synoptic-scale classifications derived for the region (e.g. type 7 is very similar to one that is associated with summertime precipitation over the southeastern USA (Diem, 2006)).' Naturally, this does not constitute a comprehensive evaluation!*

Pg5ln16 suggest adding 12 EDT /2 CDT after 20 UTC.
*Response: We have added a note regarding time zones. Thus the text now read; 'for 20:00 UTC (i.e. 16:00 Eastern Daylight Time, 15:00 Central Daylight Time)'. Please note: Eastern Daylight Time (EDT) is UTC MINUS 4 hr, not 8 hours as the reviewer wrote (maybe this is a typographic error).*

Pg5ln30 "strong [southerly] low-level advection of [high] T and Q into the region"?

*Response: Done*

Pg5ln36 "[Due to its spatial heterogeneity,] soil moisture is... "
*Response: Done*

Pg6ln3 SM plays much less of a role in the radiation-limited temperate NE forests.
*Response: Yes.*

Pg6ln6 I would not agree that MERRA-2 SM has been "extensively" evaluated- it is a very new product and only one (biased) reference is provided
*Response: We have removed 'extensively'.*

Pg6ln8 it should be specified that validation occurs only over non-forested sites, where in-situ SM is available.
*Response: Very good point – well made. We have added a caveat that reads; 'However, it should be noted that there are relatively few direct measurements of SM and thus the evaluation of MERRA-2 is focused on agricultural locations.'*

Pg6ln18 was the time difference (2PM EDT, 1PM CDT) accounted for in the comparison? If so, how?
*Response: No.*

Pg6ln19 is the GPLLJ the only source of moisture for MO and IA? Sentence is currently misleading.
*Response: Fair point we have add 'in part' to clarify.*

Pg6ln21 "and the presence of abundant SM". Intent of statement is unclear. SM is abundant in the eastern states, as well.
*Response: This is a very interesting point. There are relatively few homogeneous SM data sets. IF (and I would emphasize, if) one treats remote sensing SM estimates as equally valid over the entire study area then for our study region SM is most abundant in the region I was referring to (lower GP) and in the NE, but is less abundant in the eastern states. See figure below:*

[Figure]

*Image from: https://www.jpl.nasa.gov/spaceimages/details.php?id=PIA19337. Caption: High-resolution global soil moisture map from SMAP's combined radar and radiometer instruments, acquired between May 4 and May 11, 2015 during SMAP's commissioning phase. The map has a resolution of 5.6 miles (9 kilometers).*
*We have added a reference to (Al Bitar et al., 2017) in support of the assertion regarding abundant SM so we are not solely relying on the MERRA-2 product (and our figure 1).*

Pg6ln21 suggest list item (5) should be added, beginning from "There are also Important ..."
*Response: Done*

Pg6ln25-30 suggest moving these stats out of the data section and into Results.
*Response: We decided against doing so.*

Pg6ln30 "over [the] southern ..."
*Response: Done*

Pg7ln8 please include commentary on how these predictors were selected. What was the logic or metrics employed in the selection process?
*Response: Done (but we do so where the predictors are first introduced – i.e. in the bulleted list in section 2.2 not where the reviewer suggested).*

Pg7ln24 "readily available to [soil evaporation, as opposed to integrated soil moisture profile that constitutes the water availability to evapotranspiration]".
*Response: We have modified the sentence to read; 'Therefore, in addition to developing models using the MERRA-2 variable 'PRMC', which is the 'Total profile soil moisture content' in $m^3m^{-3}$ (that is summed across all six soil layers and represents the total water potentially available for evapotranspiration to the atmosphere), a fifth ANN model*

*(with 3 hidden layers) is also built that uses the variable 'GWETTOP' that describes the SM content in the upper 5 cm of the soil (unitless) (Reichle et al., 2017), and thus best represents the SM that is readily available for evaporation into the overlying atmosphere.'*

Pg7ln25 suggest "Table 1 summarizes the acronyms used herein for each of the five models considered. A schematic of the model architecture and data flows is provided in Fig. 3.
*Response: Done*

Pg7ln27 suggest including statement that the 70-15-15 sample subsets were identical for all model architectures.
*Response: Done*

Pg8ln17 this statement appears at odds with Table 1, which lists ANN-HL3 as having the worst model statistics.
*Response: I believe this is a misunderstanding ANN-HL3 excludes SM and indeed has highest RMSE.*

Pg9ln2 is there any statistical significance to this difference?
*Response: The reviewer is raising an interesting point – can one undertake an appropriate non-parametric test, correcting (of course) for multiplicity. We did not, instead choosing to draw out spatial consistency as a basis for inferring 'skill'.*

Pg9ln6 "where [modeled] land-atmosphere ... and where strong [longitudinal] gradients of SM"
*Response: Done*

Ph9ln13 "such [as] dry lines ..."
*Response: Done*

Pg9ln24 I question "evaporation from the Great Lakes". Is Great Lakes evaporation a strong predictor of theta-e in the eastern U.S.? Evaporation over the Great Lakes is much more substantial in the winter months and drives tremendous lake-effect snow bands, but the affected region is isolated.
*Response: Interesting point. As a resident of upstate New York I can confirm the lake-effect snows are a strong function of ice cover (and hence evaporation) from the Great Lakes! However, observationally-derived estimates of evaporation from the Great Lakes reported in (Notaro et al., 2015) are certainly 'non-trivial' (i.e. approx. an average of 45 mm in August, relative to over-lake precipitation receipt of approx. 85 mm for the period 1980-1999..). But it is an open question as to the impact on downstream theta-e, so we have added 'potentially'.*

Pg9ln30 how is water management signal being linked to the present analysis? MERRA2 has no accounting of water management.
*Response: Right, its not.*

Pg9ln32 suggest DeAngelis et al. 2010 in addition to Pryor et al., 2016 reference for warming hole
*Response: Done*

Pg10ln5 HL3-SM seems to perform the best; 3 hidden layers as opposed to 1.
*Response: Correct*

Pg10ln7 "...when all [eastern U.S.] grid cells are considered..."
*Response: Done*

Pg10ln12 please clarify that "test period" here is s*ynonymous with "independent samp*le", or 15% of all JJA days
*Response: Reworded to independent data sample.*

Pg10lns5-36. Why can't these stats on HR and FAR be added to an expanded Table 1 for each model?
*Response: This is a style consideration. Table 1 is designed to summarize the overall performance of the models (and is already quite complex), the HR/FAR naturally reflect the performance for extremes.*

Pg10ln22 statement on CLM (Buzan et al., 2015) does not appear directly relevant to MERRA-specific results being discussed here. It is out of place, given non-MERRA results are not specifically called out elsewhere in the paper. I suggest removing this statement.
*Response: It is POTENTIALLY interesting at least to some readers to note that; 'The causes of the poor model performance in eastern TX and SC are currently not fully understood, although it is worthy of note that data from MERRA-2 grid-cells in SC exhibit a relatively low overall frequency of exceedance of this threshold and are also characterized by comparatively low 99$^{th}$ percentile $\theta_e$ in an analysis of heat indices derived from the Community Land Model v4.5 (Buzan et al., 2015).'*

Pg10ln26 To include MO, IA and IL is a stretch. The central U.S. hotspot of GLACE-1 was squarely contained within 27-41N; 106-97W.
*Response: Right, the other references we cite do focus more on the eastern states. We have reworded to; 'central and eastern USA'*

Pg10ln32 please quantify "greatly" Is this statistically significant?
*Response: Fair point 'greatly' is subjective. I have deleted this word.*

Pg11ln1 "HR and FAR are comparable to (or better than) seasonal re-forecasts of summertime T at 2-m..." Please clarify were the HR and FAR of T or theta-e-max,min compared against the HR and FAR of T2m from ECMWF (ERA-I?)?
*Response: Yes. I have reworded to; 'Further, HR and FAR computed for max-$\theta_e$ and min-$\theta_e$ are comparable to (or better than) seasonal re-forecasts of summertime T at 2-m over the land areas of Southern Europe developed using the European Centre for*

*Medium-Range Weather Forecasts (ECMWF) seasonal ensemble forecasting system (Weisheimer et al., 2011).'*

Pg11ln15 the comparison between HL3-SM and HL3-TOP needs to be included and thoroughly discussed, esp. for theta-e-max.
*Response: There is a discussion which reads;' Differences in model performance between ANN conditioned on total SM and using wetness only in the top soil layer (upper 5 cm) are very small when averaged across the domain (Table 1) and indeed for virtually all grid cells. Only 26 grid cells exhibited a Δ|RMSE| > 0.5 K for models using PRMC versus those using GWETTOP (out of a total of 1962), while 155 exhibited an increase in RMSE > 0.5 K when SM was excluded from the model. Thus, although the weights within the ANNs differ for use of the two SM parameters, the overall model skill is unchanged by use of the two SM estimates possibly due to the spatial and temporal averaging applied herein, or uncertainty in reanalysis-derived SM variables.'*

Pg11ln32 "from the [MERRA-2]..."
*Response: Done*

Pg11ln33 suggest "important differences in the magnitude of derived equivalent temperature (Te)[,as well as in strength of land-atmosphere coupling between the reanalysis products (e.g., Ferguson et al., 2012; Schoof et al., 2017)]"
*Response: Done*

Pg12ln9 sentence beginning "Correlation coefficients exceed..." needs to be reworded and probably split into two or more sentences.
*Response: Done*

Pg12ln11 sentence beginning "This is true for the simulation..." is confusing. Does this imply that the prior sentence holds exactly for max theta e?
*Response: Reworded for clarity to ; 'ANN-HL3-SM models also exhibit highest skill for simulation of extreme min- and max-$\theta_e$.'*

Pg12ln15 the statement that "min theta-e exhibits a stronger dependence on the precise prevailing synoptic scale conditions" is unsupported by the analysis and accompanying discussion. Where has the link between min theta-e predictability and PC number been established and probed?
*Response: Very good point. We have re-written this sentence to read; 'Results from the ANN models further indicate that max-$\theta_e$ and the occurrence of extreme max-$\theta_e$ appear to be considerably more sensitive to SM than min-$\theta_e$ which in turn appears to exhibit a stronger dependence on the precise prevailing synoptic scale conditions based on the ANN weights.' To reflect the assessment I made of the weights.*

Pg12ln28 what are the alternative hypotheses in literature for the 'warming hole'? references here or previously upon the introduction of the term would be useful.
*Response: Done*

Pg12ln30 I disagree that the statistical modeling exercise documented herein has "enhance[d] mechanistic understanding of the causes of variability and change in theta-e". This claim should be deleted.
*Response: Deleted mechanistic.*

Pg12ln35. The authors should comment on lessons learned and insight gleaned. For example, the weighting for the global T was stated to be *negligible. Then, is their re*commendation to forgo global T requirements in future ANN pursuits? Similarly, is the total integrated column soil moisture necessary or merely the surface layer? Does the fact that MERRA2 uses bias-corrected P make any allusions of similar success in full forecast models (with biased, model P) unfair and misleading? In addition to those predictors tested here, which other predictors would be meaningful to explore/consider?
*Response: Well what we actually state is; 'However, it is worthy of note that the weights on the index of global temperatures (and thus expression of internal climate modes) in ANN-HL3-SM are uniformly close to zero across the grid-cells in the interior of the continent, but are of large magnitude in land grid cells close to the Gulf of Mexico (i.e. the south of the domain) and around the Great Lakes.' So no we would not suggest foregoing global T.*
*We have elaborated on 'next steps' and the implications of our work by adding text to the conclusions. Section 4 now read;*

[revised manuscript text omitted]

Table 1. Caption should specify over JJA and 1980-2014; "coefficient [(r)]", "the presence [or] absence"; the total number of grids (1962) should be specified OR the stats should be presented as percentages to be consistent with the text. Column with r>0.8 and RMSE <5k: what about the number of grids that satisfy BOTH criteria?
*Response: Done and done (new data added to Table 1, and a note added to the text that reads; 'The ANN-HL3-SM model also exhibits the highest number of grid cells that have both a RMSE < 5 K and a r > 0.8 for both max-$\theta_e$ and min-$\theta_e$. (Table 1).').*

Fig 1. For (c-g) it is unclear from the caption whether 1980-2014 is still the averaging period; "standard deviation of daily [JJA] (e)..."; Mean [JJA daily] soil moisture..."; explained computation would be more appropriate to include in the main text of article. Is SM also averaged over 1980-2014? ; "total profile [0-1m? profile total thickness]"; the variable names and units should be displayed on each subpanel. Is the 90-day running JJA just June1-Aug31 average or on June 1 the ~March1-June1 average and so-on? Please clarify.
*Response: Clarifications added. In the text we have added the following details about SM; 'Specifically we use a 90-day running mean estimate of antecedent SM (i.e. the value for 1 June is an average of values from approx. 1 March to 1 June) over 3×3 grid cells centered on the grid cell in question (i.e. integrated over an area of approximately 30,000 km$^2$, see Figure 1g).' This is already a relatively compact figure so have maintained the units in the caption.*

Fig3. The 5 models from Table 1 should be alluded to here by their acronyms in that table; the domain of actions on the right hand side should be specified as for the

predictors. From the flowchart is appears that z-scores are fed into the ANN. From the text I understood that daily z-scores are used to compute daily PC's which are passed to the ANN. This Figure could be eliminated by adding a data table, which demarcates the predictors and predictands, and describing the rest in text. Currently the flowchart is a bit unclear.

*Response: I've remade the figure to define the acronyms and clarify the data flows.*

Fig 4. ANN-HL3-TOP is missing and needs to be included. The labels and units for rows 2 and 3 should be added to the far right side.

*Response: ANN-HL3-TOP is qualitatively so similar to ANN-HL3-SM that in order to enhance the legibility of the figure I excluded it. I have added a note to the caption to reflect this. This is already a relatively compact figure so have maintained the units in the caption.*

References (Bolton 1980; Bryan 2008; Davies-Jones 2009; DeAngelis et al. 2010; Dirmeyer and Halder 2017; Ferguson et al. 2012; Guo et al. 2006)
*…<References supplied by the reviewer deleted for brevity> …*

References cited in this response
Al Bitar, A., Mialon, A., Kerr, Y. H., Cabot, F., Richaume, P., Jacquette, E., Quesney, A., Mahmoodi, A., Tarot, S., Parrens, M., Al-Yaari, A., Pellarin, T., Rodriguez-Fernandez, N., and Wigneron, J. P.: The global SMOS Level 3 daily soil moisture and brightness temperature maps, Earth System Science Data, 9, 293-315, 10.5194/essd-9-293-2017, 2017.
Brown, D. G., Johnson, K. M., Loveland, T. R., and Theobald, D. M.: Rural land-use trends in the conterminous United States, 1950-2000, Ecological Applications, 15, 1851-1863, 10.1890/03-5220, 2005.
Buzan, J. R., Oleson, K., and Huber, M.: Implementation and comparison of a suite of heat stress metrics within the Community Land Model version 4.5, Geoscientific Model Development, 8, 151-170, 10.5194/gmd-8-151-2015, 2015.
Davey, C. A., Pielke, R. A., Sr., and Gallo, K. P.: Differences between near-surface equivalent temperature and temperature trends for the Eastern United States - Equivalent temperature as an alternative measure of heat content, Global and Planetary Change, 54, 19-32, 10.1016/j.gloplacha.2005.11.002, 2006.
Diem, J. E.: Synoptic-scale controls of summer precipitation in the Southeastern United States, Journal of Climate, 19, 613-621, 10.1175/jcli3645.1, 2006.
Ferguson, C. R., Wood, E. F., and Vinukollu, R. K.: A Global Intercomparison of Modeled and Observed Land-Atmosphere Coupling, Journal of Hydrometeorology, 13, 749-784, 10.1175/jhm-d-11-0119.1, 2012.
Ford, T. W., and Schoof, J. T.: Oppressive Heat Events in Illinois Related to Antecedent Wet Soils, Journal of Hydrometeorology, 17, 2713-2726, 2016.
Ford, T. W., and Schoof, J. T.: Characterizing extreme and oppressive heat waves in Illinois, Journal of Geophysical Research: Atmospheres, 122, 682-698, 2017.
Koster, R. D., Sud, Y., Guo, Z., Dirmeyer, P. A., Bonan, G., Oleson, K. W., Chan, E., Verseghy, D., Cox, P., and Davies, H.: GLACE: the global land–atmosphere coupling experiment. Part I: overview, Journal of Hydrometeorology, 7, 590-610, 2006.

[revised manuscript text omitted]

---

## Author Comment (AC2) · 1 Nov 2017

Response to: Interactive comment on "Modeling the contributions of global air temperature, synoptic-scale phenomena and soil moisture to near-surface static energy variability using artificial neural networks" by Sara C. Pryor et al. Anonymous Referee #3

Below we list the comments of the reviewer and our responses (in italics below each point). At the end of this document we also provide a tracked changes version of

the manuscript showing the changes we have made in full. A pdf of this entire file is provided as a supplement to this response.

...<Preamble deleted for brevity> ... QUOTE START: I think this work will make a valuable contribution to understand the drivers of surface static energy and heat waves, and would recommend publication after some minor revisions.

My comments are listed below: 1) There major drivers (e.g., global temperature, synoptic-scale indices and SM) were used to develop statistical models for $\theta$e. By comparing the statistical models with and without considering the SM (e.g., ANN-HL3-SM vs ANN-HL3), the authors concluded that the SM played a key role in dictating the warming holes. This statement may be misleading. To identify the relative roles of individual drivers, it might be better to develop statistical models by examining different combinations of the drivers (e.g., global temperature and SM, or synoptic-scale indices and SM). It is also possible that the roles of different drivers may vary in different regions. Response: Yes, quite. We do not infer that a key role of SM is generalizable across land areas ... we have modified our statement in the conclusions to read; 'Our results imply there are large spatial gradients in the importance of the predictors we used herein and also that our framework has greater skill for max-$\theta$e than min-$\theta$e. It is also important to note that in the northeastern portions of our study region inclusion of SM as a predictor has considerably lower impact on model skill for either max-$\theta$e or min-$\theta$e (Figure 4-7). Global T substantially contributes to model skill near the Gulf coast and close to the Great Lakes but is less important over the remainder to the eastern USA, while SM exhibits greatest importance in sub-regions previously noted as exhibiting 'warming holes'. It is possible that inclusion of additional predictors could lead to enhanced model skill particularly for extreme high values of max-$\theta$e or min-$\theta$e that are of greatest importance to human health, and/or derivation of persistence indices (e.g. the occurrence of consecutive nights with high minimum $\theta$e). We can not conclusively discount contributions from other phenomena (e.g. aerosol forcing, cloud cover) to the occurrence of 'warming holes' (areas with declining or no-trends in T) (Meehl et al.,

2015), and these features may be a complex response to multiple drivers. However, results presented herein are consistent with past work that has indicated the importance of soil moisture (SM) in determining partitioning of the surface energy budget, and thus the spatiotemporal patterns of $\theta e$ over the central and eastern USA (Koster et al., 2011;Koster et al., 2006;Pryor and Schoof, 2016;Pryor et al., 2016;Ford and Schoof, 2016, 2017;McKinnon et al., 2016). Indeed, SM is particularly important in determining the surface energy partitioning and the magnitude of $\theta e$ over regions that have previously been identified as exhibiting 'warming holes', and for all grid cells the RMSE for models including SM as a predictor is smaller than the temporal variability of $\theta e$ as measured using the standard deviation of the daily $\theta e$ values. Specifically, only a model including SM is able to predict the occurrence of extreme (and highly health-relevant) values of $\theta e$ over the western portion of Midwestern states such as IA, MO, IL and also in MS and AL. This research thus implies that SM has played and may continue to play a key role in dictating the presence and intensity of 'warming holes' that have been previously noted in analyses of near-surface air temperature data (from both in situ measurements and reanalysis products).' As the reviewer suggests, an alternative approach to the methodology we employed is a stepwise MLR but of course that would mean apriori assumptions regarding the term interactions. But we hope our research will inspire other studies that could employ the approach the reviewer proposes.

2) The climate variations in the eastern USA are influenced by different climate modes (e.g., ENSO, NAO, IPO). The aerosol may also play some roles on the warming holes. It is not clear why these modes were not used in this study. Response: Yes, the reviewer is quite correct – as we state in the introduction; 'Global scale forcing due to enhanced greenhouse gas concentrations and internal climate variability (e.g. ENSO (Mann et al., 1998)). As T increases the atmospheric water vapor content responds in proportion to the saturation vapor pressure (Willett et al., 2007;Allen and Ingram, 2002). Thus, both components (T and q) of static energy ($\theta e$) are enhanced in years and seasons with high global mean air temperatures. Previous research has indicated that variability in equivalent temperature (Te, i.e. the temperature computed from (1) but

excluding the correction for bringing the air parcel adiabatically to a reference pressure of 1000 hPa) in the North Atlantic is strongly linked to NAO (Ribera et al., 2004), and the probability of 'heat waves' across the US is linked to hemispheric waves (Teng et al., 2013) and thus the PNA (Trenberth, 1990).' We did not explicitly include the teleconnection indices because they will influence the synoptic PC scores. And yes, we quite concur, aerosol radiative forcing may indeed play a key role in warming holes. . . we have modified our statement in the conclusions to read; 'We can not conclusively discount contributions from other phenomena (e.g. aerosol forcing, cloud cover) to the occurrence of 'warming holes' (areas with declining or no-trends in T) (Meehl et al., 2015), and these features may be a complex response to multiple drivers.'

3) For SM index. The 90-day running mean estimate of antecedent SM in 3x3 grid cells were used. Why it is necessary use the 90-day running mean and 3x3 grid cells average? It was found that the SM would influence the climate downstream. Do you think it is possible to improve the model results by averaging the SM over a large region (e.g., 5x5 or 10x10 grid cells)? Response: Yes, this is an interesting point. We have elaborated on 'next steps' by adding; 'Thus, there would be value in applying this framework to additional observationally constrained data sets to evaluate: (1) The degree to which the findings of a key role of SM to determining the model skill for daily maximum $\theta$e in specific sub-regions are generalizable and spatially consistent between reanalyses, and further if the predictability of $\theta$e exhibits sensitivity to the spatiotemporal averaging used in deriving the SM predictors. (2) If use of a reanalysis product (or forecast model) that does not employ bias-correction of precipitation amounts would substantially alter the ANN model structure. (3) If the partial truncation of the upper percentiles of daily maximum $\theta$e in the model predictions is also a generalizable finding when our model framework is applied to different data sets.'

4) Page 4, line 15. How were the daily maximum and minimum $\theta$ e calculated? Did you firstly compute the $\theta$e using the hourly T, q, and P, and then derive the maximum and minimum values? Response: Yes. We have clarified by writing; 'An estimate of (i) daily

maximum $\theta$e and (ii) daily minimum $\theta$e in each grid cell (see Figure 1c and d). The daily minimum and maximum values are used as the predictands in the downscaling and are derived using Eq(1) applied to hourly T at 2-m, q at 2-m and surface pressure (P).'

5) Page 5, last sentence. The global T and 15 synoptic-scale PC scores are common to models built for all grid cells, whereas the SM is grid-cell specific. Therefore, it is not a surprise that the SM plays a more important role in the model performance. Response: I am not sure why this would be the case since the models are built grid cell by grid cell and thus the ANN (and MLR) weights can vary in space – thus I am not sure an a priori case could be made that SM must per se exhibit a stronger influence.

6) Table 1. The # of grid cells with r>0.8 & RMSE < 5K were shown. However, it might be better to show the percentage because the number of grid cells depends on the spatial resolution of the dataset used. Response: Yes, I have given the total number of grid cells in the caption to help contextualize the numbers.

7) Figure 2. Would it possible to add the state or continental boundary to the figures? Response: I tried a number of different iterations attempting to add additional lines but sadly it rendered the frames less and less legible.

References cited in this response Allen, M., and Ingram, W.: Constraints on future changes in climate and the hydrologic cycle, Nature, 419, 224-232, 2002. Ford, T. W., and Schoof, J. T.: Oppressive Heat Events in Illinois Related to Antecedent Wet Soils, Journal of Hydrometeorology, 17, 2713-2726, 2016. Ford, T. W., and Schoof, J. T.: Characterizing extreme and oppressive heat waves in Illinois, Journal of Geophysical Research: Atmospheres, 122, 682-698, 2017. Koster, R. D., Sud, Y., Guo, Z., Dirmeyer, P. A., Bonan, G., Oleson, K. W., Chan, E., Verseghy, D., Cox, P., and Davies, H.: GLACE: the global land–atmosphere coupling experiment. Part I: overview, Journal of Hydrometeorology, 7, 590-610, 2006. Koster, R. D., Mahanama, S., Yamada, T., Balsamo, G., Berg, A., Boisserie, M., Dirmeyer, P., Doblas-Reyes,

F., Drewitt, G., and Gordon, C.: The second phase of the global land–atmosphere coupling experiment: soil moisture contributions to subseasonal forecast skill, Journal of Hydrometeorology, 12, 805-822, 2011. Mann, M. E., Bradley, R. S., and Hughes, M. K.: Global-scale temperature patterns and climate forcing over the past six centuries, Nature, 392, 779-787, 1998. McKinnon, K. A., Rhines, A., Tingley, M. P., and Huybers, P.: Long-lead predictions of eastern United States hot days from Pacific sea surface temperatures, Nature Geoscience, 9, 389-394, 10.1038/ngeo2687, 2016. Meehl, G. A., Arblaster, J. M., and Chung, C. T. Y.: Disappearance of the southeast US "warming hole" with the late 1990s transition of the Interdecadal Pacific Oscillation, Geophysical Research Letters, 42, 5564-5570, 10.1002/2015gl064586, 2015. Pryor, S. C., and Schoof, J. T.: Evaluation of near‐surface temperature, humidity, and equivalent temperature from regional climate models applied in type II downscaling, Journal of Geophysical Research: Atmospheres, 121, 3326-3338, 2016. Pryor, S. C., Sullivan, R. C., and Wright, T.: Quantifying the roles of changing albedo, emissivity, and energy partitioning in the impact of irrigation on atmospheric heat content, Journal of Applied Meteorology and Climatology, 55, 1699-1706, 2016. Ribera, P., Gallego, D., Gimeno, L., Perez-Campos, J., García-Herrera, R., Hernández, E., de La Torre, L., Nieto, R., and Calvo, N.: The use of equivalent temperature to analyse climate variability, Studia Geophysica et Geodaetica, 48, 459-468, 2004. Teng, H., Branstator, G., Wang, H., Meehl, G. A., and Washington, W. M.: Probability of US heat waves affected by a subseasonal planetary wave pattern, Nature Geoscience, 6, 1056-1061, 2013. Trenberth, K. E.: Recent observed interdecadal climate changes in the Northern Hemisphere, Bulletin of the American Meteorological Society, 71, 988-993, 1990. Willett, K. M., Gillett, N. P., Jones, P. D., and Thorne, P. W.: Attribution of observed surface humidity changes to human influence, Nature, 449, 710-712, 2007.

Please also note the supplement to this comment:
https://www.atmos-chem-phys-discuss.net/acp-2017-367/acp-2017-367-AC2-supplement.pdf

[Figure]

**Supplement:**

Below we list the comments of the reviewer and our responses (in italics below each point). At the end of this document we also provide a tracked changes version of the manuscript showing the changes we have made in full.

…*<Preamble deleted for brevity>* … QUOTE START: I think this work will make a valuable contribution to understand the drivers of surface static energy and heat waves, and would recommend publication after some minor revisions.

My comments are listed below:
1) There major drivers (e.g., global temperature, synoptic-scale indices and SM) were used to develop statistical models for θe. By comparing the statistical models with and without considering the SM (e.g., ANN-HL3-SM vs ANN-HL3), the authors concluded that the SM played a key role in dictating the warming holes. This statement may be misleading. To identify the relative roles of individual drivers, it might be better to develop statistical models by examining different combinations of the drivers (e.g., global temperature and SM, or synoptic-scale indices and SM). It is also possible that the roles of different drivers may vary in different regions.
*Response: Yes, quite. We do not infer that a key role of SM is generalizable across land areas … we have modified our statement in the conclusions to read; 'Our results imply there are large spatial gradients in the importance of the predictors we used herein and also that our framework has greater skill for max-$\theta_e$ than min-$\theta_e$. It is also important to note that in the northeastern portions of our study region inclusion of SM as a predictor has considerably lower impact on model skill for either max-$\theta_e$ or min-$\theta_e$ (Figure 4-7). Global T substantially contributes to model skill near the Gulf coast and close to the Great Lakes but is less important over the remainder to the eastern USA, while SM exhibits greatest importance in sub-regions previously noted as exhibiting 'warming holes'. It is possible that inclusion of additional predictors could lead to enhanced model skill particularly for extreme high values of max-$\theta_e$ or min-$\theta_e$ that are of greatest importance to human health, and/or derivation of persistence indices (e.g. the occurrence of consecutive nights with high minimum $\theta_e$).*
*We can not conclusively discount contributions from other phenomena (e.g. aerosol forcing, cloud cover) to the occurrence of 'warming holes' (areas with declining or no-trends in T) (Meehl et al., 2015), and these features may be a complex response to multiple drivers. However, results presented herein are consistent with past work that has indicated the importance of soil moisture (SM) in determining partitioning of the surface energy budget, and thus the spatiotemporal patterns of $\theta_e$ over the central and eastern USA (Koster et al., 2011;Koster et al., 2006;Pryor and Schoof, 2016;Pryor et al., 2016;Ford and Schoof, 2016, 2017;McKinnon et al., 2016). Indeed, SM is particularly important in determining the surface energy partitioning and the magnitude of $\theta_e$ over regions that have previously been identified as exhibiting 'warming holes', and for all*

*grid cells the RMSE for models including* SM *as a predictor is smaller than the temporal variability of* $\theta_e$ *as measured using the standard deviation of the daily* $\theta_e$ *values. Specifically, only a model including* SM *is able to predict the occurrence of extreme (and highly health-relevant) values of* $\theta_e$ *over the western portion of Midwestern states such as IA, MO, IL and also in MS and AL. This research thus implies that* SM *has played and may continue to play a key role in dictating the presence and intensity of 'warming holes' that have been previously noted in analyses of near-surface air temperature data (from both in situ measurements and reanalysis products).' As the reviewer suggests, an alternative approach to the methodology we employed is a stepwise MLR but of course that would mean apriori assumptions regarding the term interactions. But we hope our research will inspire other studies that could employ the approach the reviewer proposes.*

2) The climate variations in the eastern USA are influenced by different climate modes (e.g., ENSO, NAO, IPO). The aerosol may also play some roles on the warming holes. It is not clear why these modes were not used in this study.

*Response: Yes, the reviewer is quite correct – as we state in the introduction; 'Global scale forcing due to enhanced greenhouse gas concentrations and internal climate variability (e.g. ENSO (Mann et al., 1998)). As T increases the atmospheric water vapor content responds in proportion to the saturation vapor pressure (Willett et al., 2007;Allen and Ingram, 2002). Thus, both components (T and q) of static energy ($\theta_e$) are enhanced in years and seasons with high global mean air temperatures. Previous research has indicated that variability in equivalent temperature ($T_e$, i.e. the temperature computed from (1) but excluding the correction for bringing the air parcel adiabatically to a reference pressure of 1000 hPa) in the North Atlantic is strongly linked to NAO (Ribera et al., 2004), and the probability of 'heat waves' across the US is linked to hemispheric waves (Teng et al., 2013) and thus the PNA (Trenberth, 1990).' We did not explicitly include the teleconnection indices because they will influence the synoptic PC scores. And yes, we quite concur, aerosol radiative forcing may indeed play a key role in warming holes… we have modified our statement in the conclusions to read; 'We can not conclusively discount contributions from other phenomena (e.g. aerosol forcing, cloud cover) to the occurrence of 'warming holes' (areas with declining or no-trends in T) (Meehl et al., 2015), and these features may be a complex response to multiple drivers.'*

3) For SM index. The 90-day running mean estimate of antecedent SM in 3x3 grid cells were used. Why it is necessary use the 90-day running mean and 3x3 grid cells average? It was found that the SM would influence the climate downstream. Do you think it is possible to improve the model results by averaging the SM over a large region (e.g., 5x5 or 10x10 grid cells)?

*Response: Yes, this is an interesting point. We have elaborated on 'next steps' by adding; 'Thus, there would be value in applying this framework to additional observationally constrained data sets to evaluate: (1) The degree to which the findings of a key role of SM to determining the model skill for daily maximum $\theta_e$ in specific sub-regions are generalizable and spatially consistent between reanalyses, and further if the predictability of $\theta_e$ exhibits sensitivity to the spatiotemporal averaging used in deriving*

*the SM predictors. (2) If use of a reanalysis product (or forecast model) that does not employ bias-correction of precipitation amounts would substantially alter the ANN model structure. (3) If the partial truncation of the upper percentiles of daily maximum $\theta_e$ in the model predictions is also a generalizable finding when our model framework is applied to different data sets.'*

4) Page 4, line 15. How were the daily maximum and minimum θ e calculated? Did you firstly compute the θe using the hourly T, q, and P, and then derive the maximum and minimum values?
*Response: Yes. We have clarified by writing; 'An estimate of (i) daily maximum $\theta_e$ and (ii) daily minimum $\theta_e$ in each grid cell (see Figure 1c and d). The daily minimum and maximum values are used as the predictands in the downscaling and are derived using Eq(1) applied to hourly T at 2-m, q at 2-m and surface pressure (P).'*

5) Page 5, last sentence. The global T and 15 synoptic-scale PC scores are common to models built for all grid cells, whereas the SM is grid-cell specific. Therefore, it is not a surprise that the SM plays a more important role in the model performance.
*Response: I am not sure why this would be the case since the models are built grid cell by grid cell and thus the ANN (and MLR) weights can vary in space – thus I am not sure an a priori case could be made that SM must per se exhibit a stronger influence.*

6) Table 1. The # of grid cells with r>0.8 & RMSE < 5K were shown. However, it might be better to show the percentage because the number of grid cells depends on the spatial resolution of the dataset used.
*Response: Yes, I have given the total number of grid cells in the caption to help contextualize the numbers.*

7) Figure 2. Would it possible to add the state or continental boundary to the figures?
*Response: I tried a number of different iterations attempting to add additional lines but sadly it rendered the frames less and less legible.*

References cited in this response
Allen, M., and Ingram, W.: Constraints on future changes in climate and the hydrologic cycle, Nature, 419, 224-232, 2002.
Ford, T. W., and Schoof, J. T.: Oppressive Heat Events in Illinois Related to Antecedent Wet Soils, Journal of Hydrometeorology, 17, 2713-2726, 2016.
Ford, T. W., and Schoof, J. T.: Characterizing extreme and oppressive heat waves in Illinois, Journal of Geophysical Research: Atmospheres, 122, 682-698, 2017.
Koster, R. D., Sud, Y., Guo, Z., Dirmeyer, P. A., Bonan, G., Oleson, K. W., Chan, E., Verseghy, D., Cox, P., and Davies, H.: GLACE: the global land–atmosphere coupling experiment. Part I: overview, Journal of Hydrometeorology, 7, 590-610, 2006.
Koster, R. D., Mahanama, S., Yamada, T., Balsamo, G., Berg, A., Boisserie, M., Dirmeyer, P., Doblas-Reyes, F., Drewitt, G., and Gordon, C.: The second phase of the global land–atmosphere coupling experiment: soil moisture contributions to subseasonal forecast skill, Journal of Hydrometeorology, 12, 805-822, 2011.

[revised manuscript text omitted]

5) There are also important spatial patterns of the day-to-day variability of daily max-$\theta_e$ and min-$\theta_e$ that provide key context for considering the performance of different transfer functions (i.e. statistical models linking indices of the global mean $T$, synoptic scale meteorology and soil moisture (the predictors) to the response variables; max-$\theta_e$ and min-$\theta_e$). The variance (and standard deviation) of daily max-$\theta_e$ values (i.e. a measure of the dispersion of individual days around the mean values of min- and max-$\theta_e$ shown in Figures 1c and 1d) is largest over IA/southern Minnesota (MN) (Figure 1e), while the variance of min-$\theta_e$ is greatest over Illinois (IL) (Figure 1f). In both cases the day-to-day variability as measured by the standard deviation is ~ 10 K. The standard deviation of daily max-$\theta_e$ and min-$\theta_e$ decreases with decreasing latitude and both variables exhibit lowest variability over the southern portions of the study domain (e.g. Florida (FL) has a standard deviation < 5 K) (Figure 1e,f).

Thus the time series of predictors 1) and 2) (global mean $T$ and the 15 PC scores) are common to models built for all grid cells, but predictor 3) (*SM*) and the response variables (predictands, daily max-$\theta_e$ and min-$\theta_e$) are grid-cell specific.

**2.3 Methods**

Artificial neural network (ANN) architectures are potentially highly useful in developing statistical models for response variables such as $\theta_e$ because ANN do not require any assumptions about the form of the relationship between individual predictors, between predictors and predictands (min- and max-$\theta_e$), and can treat complex and non-linear term interactions. ANNs are data-driven self-adaptive multi-layer perceptrons that model relationships between input variables and dependent output variables. Term interactions are described using mathematical

Prof. Sara C. Pryor 10/30/2017 4:50 PM

Prof. Sara C. Pryor 10/31/2017 12:35 PM

Prof. Sara C. Pryor 10/30/2017 4:52 PM

Prof. Sara C. Pryor 10/30/2017 5:10 PM

Prof. Sara C. Pryor 10/31/2017 12:01 PM

Prof. Sara C. Pryor 10/31/2017 12:01 PM

[revised manuscript text omitted]

Prof. Sara C. Pryor 10/30/2017 6:19 PM

Prof. Sara C. Pryor 11/1/2017 10:49 AM

Prof. Sara C. Pryor 11/1/2017 10:50 AM

[Figure]

Figure 1. (a) The study domain used herein (25.5-50°N, 97.5-65°W) and five consolidated land use land-cover (LULC) classes as represented at a resolution of 0.05×0.05° from the MODIS land cover data set for 2014 (MCD12C). (b) A boxplot of daily global mean JJA temperature values (K) (1980-2015) as computed from the MERRA-2 output. (c) Mean summertime (JJA, 1980-2015) maximum-$\theta_e$ and (d) minimum-$\theta_e$ (K) as computed from hourly $T$ and $q$ at 2-m and surface pressure ($P$) in each MERRA-2 grid cell. Standard deviation of daily (e) maximum-$\theta_e$ and (f) minimum-$\theta_e$ (K) (JJA, 1980-2015). (g) Mean soil moisture estimates (1980-2015) from MERRA-2 used as a predictor in the ANN and computed as follows: the daily mean value for total profile soil moisture content (PRMC) for each grid cell is averaged over a spatial area of 3×3 grid cells centered on the grid cell of interest, and used to compute a 90-day running JJA mean soil moisture ($m^3 m^{-3}$). Panel (h) shows the location of states that are explicitly referred to herein.

Prof. Sara C. Pryor 11/1/2017 11:29 AM

[revised manuscript text omitted]